# Efficient use of cement and concrete to reduce reliance on supply-side technologies for net-zero emissions

Takuma Watari [1,2 ✉], Zhi Cao[3 ✉], Sho Hata [1,4] & Keisuke Nansai [1]

Decarbonization strategies for the cement and concrete sector have relied heavily on supply-side technologies, including carbon capture and storage (CCS), masking opportunities for demand-side intervention. Here we show that cross-cutting strategies involving both the supply and demand sides can achieve net-zero emissions by 2050 across the entire Japanese cement and concrete cycle without resorting to mass deployment of CCS. Our analysis shows that a series of mitigation efforts on the supply side can reduce 2050 $CO_2$ emissions by up to 80% from baseline levels and that the remaining 20% mitigation gap can be fully bridged by the efficient use of cement and concrete in the built environment. However, this decarbonization pathway is dependent on how $CO_2$ uptake by carbonation and carbon capture and utilization is accounted for in the inventory. Our analysis underscores the importance of including demand-side interventions at the heart of decarbonization strategies and highlights the urgent need to discuss how to account for $CO_2$ uptake in national inventories under the Paris Agreement.

[1] Material Cycles Division, National Institute for Environmental Studies, Tsukuba, Japan. [2] Institute for Sustainable Futures, University of Technology Sydney, Sydney, NSW, Australia. [3] Energy and Materials in Infrastructure and Buildings (EMIB), University of Antwerp, Antwerp, Belgium. [4] Graduate School of Frontier Sciences, The University of Tokyo, Kashiwa, Japan. ✉email: watari.takuma@nies.go.jp; zhi.cao@uantwerpen.be

Compliance with internationally agreed upon climate targets depends to a great extent on how to address hard-to-abate industrial sectors[1]. A prime example is concrete, the most extensively used human-made material in the world[2]. The chemical reactions and high temperatures required for the production of cement, the binding agent for making concrete, combined with its mass production and use, make its decarbonization one of our most significant challenges[3,4]. The share of $CO_2$ emissions arising from the entire cement and concrete cycle in total global energy-related emissions has been increasing in recent decades[5], and is now ~10%[6,7]. As scientific knowledge regarding climate change advances[8], the key question centers on what actions need to be taken, on what scale, and by when, in order to decarbonize the complete cement and concrete cycle.

A growing body of evidence has shown the potential for emissions reduction through efforts focusing on the supply side of the cement and concrete cycle, including energy efficiency improvements[7,9], clinker-to-cement ratio reduction[10,11], low-carbon fuel utilization[12], cement substitution with alternative binders[13,14], carbon capture and storage (CCS)[15,16], and carbon capture and utilization (CCU)[17]. Among these various supply-side measures, industry pledges and policy discussions are particularly dependent on CCS[18–20], even though it belongs to the lowest hierarchy of strategies due to technology lock-in concerns and low resource efficiency[21]. In contrast, an emerging research stream has shown that significant, but largely untapped, opportunities exist on the demand side through more efficient material use[22–26]. However, these studies do not provide a pathway for achieving net-zero emissions by approximately mid-century, an underlying requirement for meeting the 1.5–2 °C climate target[8]. Recent literature reviews summarizing the large body of available evidence suggest the importance of cross-cutting strategies that span the entire cement and concrete cycle[21,27–29]; however, relevant empirical analyses remain largely lacking. While several pioneering efforts have been undertaken to fill this gap[6,30], they tend to consider only a limited set of concrete end uses and thus do not capture a large part of concrete flows and stocks.

The work presented here addresses these knowledge gaps by characterizing the net-zero emission pathway for the cement and concrete cycle using a cohesive modeling framework. Our approach builds on a dynamic material flow analysis (MFA) model, coupled with an emissions model that tracks $CO_2$ emissions arising from each process, and a physicochemical model that quantifies the $CO_2$ uptake associated with concrete carbonation. This integrated approach allows us to explore strategies to achieve net-zero emissions across the cement and concrete cycle while ensuring a dynamic mass balance of stocks and flows in the system. We apply the model to Japan given its significant $CO_2$ emissions (fifth largest in the world[31]) and the availability of the types of detailed datasets that are required for such a modeling approach. Japan is one of the few countries globally that provides government-managed, rigorous, and long-term data for construction activity by structure type and material intensity. This study first scrutinizes the historical structure and evolution of the cement and concrete cycle and its associated-$CO_2$ emissions and uptake over a 70-year period, from 1950 to 2019. The proposed model is then used to explore 16 strategies (nine supply-side and seven demand-side interventions) for achieving net-zero emissions by 2050. The principal storyline here focuses on whether it is possible to achieve net-zero emissions across the cement and concrete cycle without relying on CCS, which belongs to the lowest tier of mitigation strategies[21].

## Results

**Contemporary cement and concrete cycle.** The detailed map of the contemporary cement and concrete cycle and associated-$CO_2$ fluxes shown in Fig. 1 offers insights into intervention

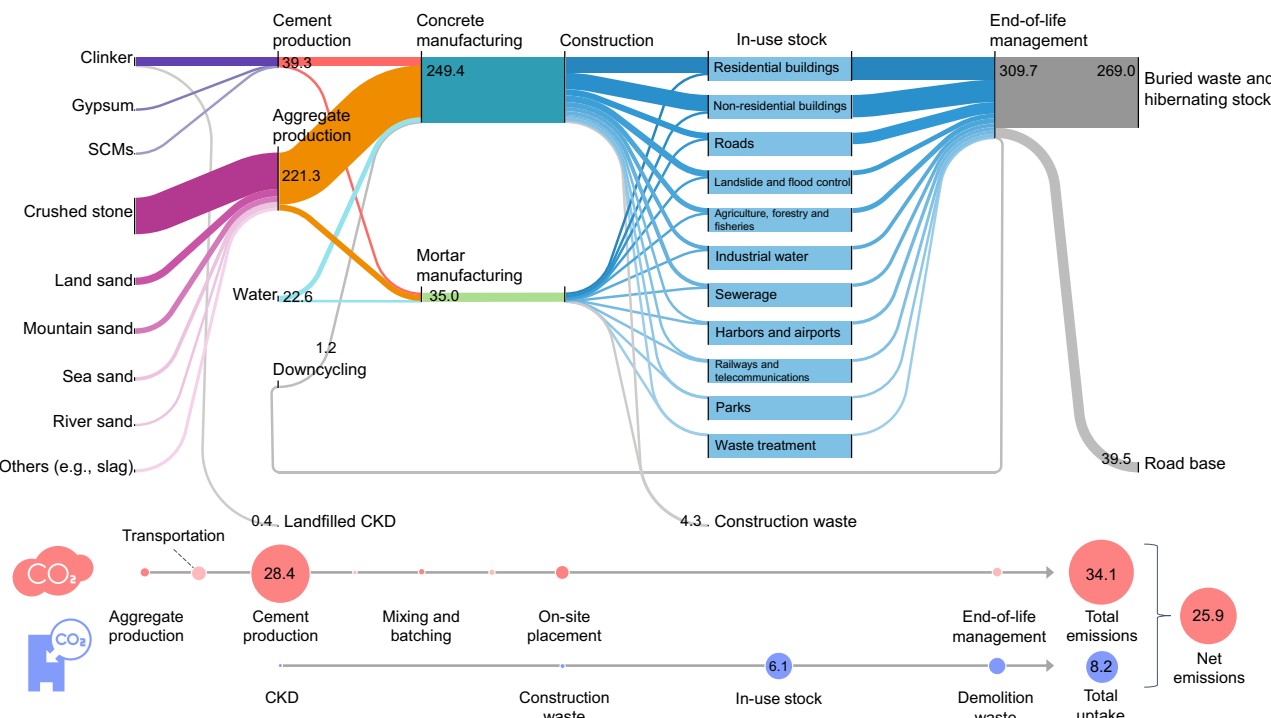

**Fig. 1 Cement and concrete cycle and associated-CO₂ fluxes in Japan in 2019.** All flows are shown to scale in Mt/year; in-use stocks shown in the box are scaled differently than flows. The numbers in the circles at the bottom of the figure show the annual $CO_2$ emissions and uptake associated with the concrete cycle in Mt-$CO_2$/year. The Sankey diagram was designed with floWeaver[63]. SCMs supplementary cementitious materials, CKD cement kiln dust.

opportunities for emission mitigation. In 2019, 284 million metric tons (Mt) of concrete and mortar (hereinafter referred to as concrete, for simplicity) were produced to meet the Japanese domestic demand for buildings and infrastructure, consisting of 14% cement, 8% batching water, 78% virgin aggregates, and less than 1% recycled aggregates. Roughly 83% of the cement weight was comprised of clinker, with the remaining 17% comprised of gypsum and supplementary cementitious materials (SCMs), suggesting a slightly higher clinker-to-cement ratio compared to other regions, including Europe and the United States[32]. Around half of the concrete produced was used for constructing buildings; the other half was used for developing infrastructure. Approximately 2% of concrete was lost as construction waste, which represents concrete that was produced but never included in a final product. In Japan, most demolished concrete is downcycled as road base materials, turned into hibernating stock, or landfilled; thus, there is an almost invisible recycling flow to concrete aggregates.

The entire cement and concrete cycle induces 34.1 Mt-$CO_2$, which is equivalent to ~3% of Japan's total $CO_2$ emissions. Of this amount, cement production accounts for the largest share (83%), followed by transportation (9%) and on-site concrete placement (5%). Despite a large production volume, the share of $CO_2$ emissions from aggregate production is only 2%. The $CO_2$ uptake by concrete carbonation reached 8.2 Mt-$CO_2$ (7.1-10.0 Mt-$CO_2$ interquartile range) in 2019, equivalent to about 24% of concrete-related emissions and ~1% of Japan's total emissions. In-use stocks act as the largest sink during the service life of buildings and/or infrastructure, absorbing 74% of the total uptake. Another 22% is derived from demolition waste at the end-of-life stage, and the remaining 4% from cement kiln dust and construction waste (Supplementary Fig. 26).

**Final demand drivers and stock dynamics**. A closer look at the demand-side drivers of the cement and concrete cycle shows that the main driving force is currently household consumption, given its high dependence on fixed capital (Fig. 2). Specifically, ~80% of cement flows are driven by household consumption, with government expenditure and exports accounting for the remainder in almost equal quantities. A more detailed breakdown of household

consumption reveals that housing is the largest driver (32%), followed by medical and health care services (10%), transportation (9%), and education (8%). "Other services", which collectively comprises retail, leisure, and so forth, accounts for 12%, suggesting that it plays a significant role in driving the current cement and concrete cycle. These trends reflect the capital-intensive nature of services: educational services require schools, medical and health care services require hospitals and nursing homes, and retail services require commercial buildings.

These final demands are actually met not by the material flows themselves, but by the stock accumulated in the society in the form of buildings and infrastructure (i.e., in-use stock). Therefore, the long-term evolutionary pattern of in-use stock provides insights into the future development path of the cement and concrete cycle. Historically, in-use concrete stock continued to increase throughout the twentieth century, reaching ~15 Gt, or 115 t per capita, in 2000 (Supplementary Fig. 21). However, its growth has stabilized since the beginning of the twenty-first century, and has generally remained constant at ~17 Gt or 130 t per capita for approximately the last 10 years. This reflects the fact that the buildings and infrastructure that are required to maintain Japan's high living standards have already been sufficiently established (Supplementary Figs. 22 and 23). Such a trend is not unique to Japan, but is common to many high-income countries (Supplementary Fig. 24). These trends provide a solid foundation for exploring how the cement and concrete cycle may evolve in the future.

**Mitigation potential of supply-side strategies**. Based on the observed historical trends, we estimated the future evolution patterns of the cement and concrete cycle and associated-$CO_2$ fluxes through 2050 (Fig. 3). Under the baseline scenario in which no mitigation strategies are implemented, the trends of $CO_2$ emissions and uptake do not change significantly over the next decades, resulting in a net emission balance of 19.8 Mt-$CO_2$ in 2050. This is equivalent to ~2% of Japan's current total $CO_2$ emissions, and is far from the net-zero emission target.

Since most of the $CO_2$ emissions associated with the cement and concrete cycle are dominated by limestone calcination and fuel combustion for heat supply, the emission saving from

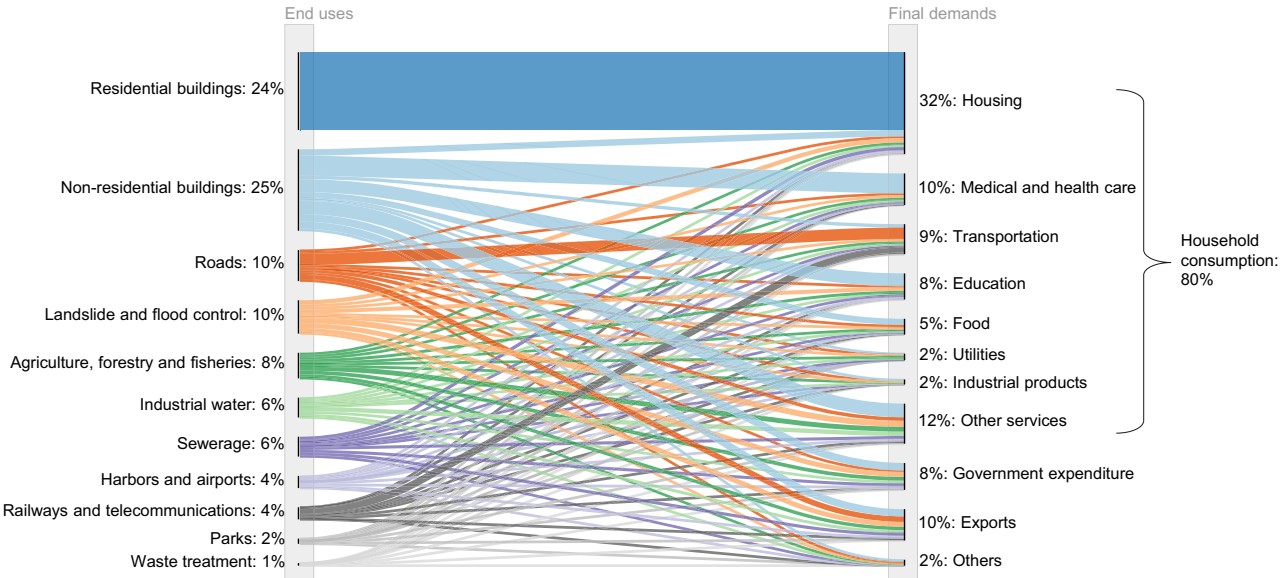

**Fig. 2 Final demand drivers of cement flows in Japan in 2019.** Due to data constraints, cement flow data as a proxy for concrete was connected to final demand drivers using the latest Japanese input-output table[52]. Here, fixed capital is endogenized by using an augmentation method[55] with the fixed capital formation table in the supplementary material of the Japanese input-output table.

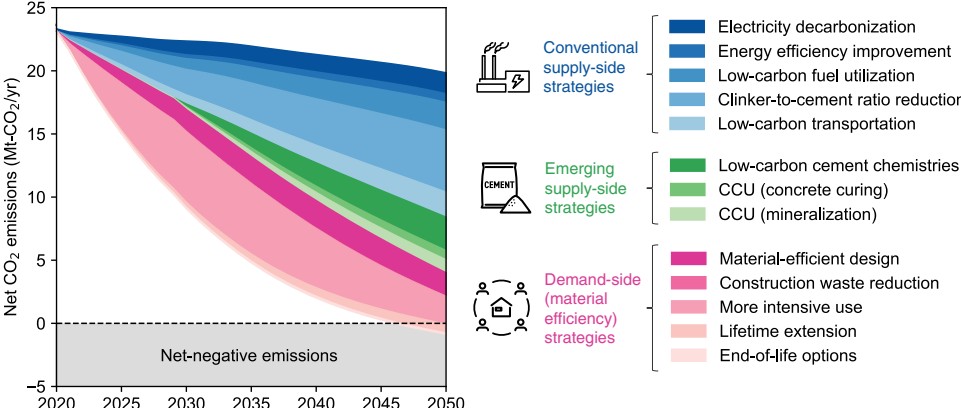

**Fig. 3 Role of supply- and demand-side strategies in net CO$_2$ emissions associated with the cement and concrete cycle in Japan, 2020–2050.** Conventional supply-side strategies are envisioned to enhance the current cement and concrete industry's efforts to reduce CO$_2$ emissions, while emerging supply-side strategies refer to strategies that are innovative, but could take some time to scale up. Demand-side strategies refer to interventions by systems and actors that use concrete in the built environment. CCU carbon capture and utilization.

electricity decarbonization is limited to 8% in 2050. Furthermore, reflecting the fact that the thermal and electrical energy efficiency of Japanese cement facilities is already close to the global 10% best-in-class, energy efficiency improvements would mitigate emissions by only 3% in 2050. In actuality, what would truly be effective is fuel switching, clinker-to-cement ratio reduction, and low-carbon transportation. Switching from the coal used in cement kilns to low-carbon fuels and reducing the clinker-to-cement ratio by increasing the use of SCMs could provide additional mitigations of 11% and 25% in 2050, respectively. Furthermore, systemic improvements in operations and logistics across all aspects of road freight, increased vehicle efficiency, and support for the use of alternative fuels in transport could result in additional savings of 10% in 2050.

Further mitigation could be achieved by several emerging strategies that are currently under active development in the cement and concrete industry. We estimate that additional emissions reductions of 13% and 7% could be achieved by 2050 through maximizing the use of low-carbon cement chemistries and CCU technologies (both concrete curing and mineralization), respectively. Importantly, however, even if all of these measures are implemented, the collective mitigation potential is only ~80%, which is 20% short of achieving net-zero emissions. This reflects the fact that emerging supply-side strategies are limited in their potential by the applicable concrete strength class and the availability of industrial waste, suggesting the need for complementary measures.

**Mitigation potential of demand-side strategies**. The remaining 20% mitigation gap could be filled by more efficient use of cement and concrete in the built environment through multi-stakeholder actions. Material-efficient design, including performance-based concrete design, use of precast concrete components, post-tensioning, and avoidance of over-design, could bring about additional emissions mitigation of 9% by 2050. Another 11% mitigation could be achieved through more intensive use of buildings and infrastructure, through such means as enhanced sharing practices and consolidation of urban functions. These efforts, together with the supply-side strategies, put us within reach of net-zero emissions. Extending the service life of buildings and infrastructure further provides additional mitigation of 4%, enabling even deeper and faster decarbonization. Construction waste reduction, component reuse, downcycling, and stockpiling of demolition waste together all play a limited role in emissions reduction, but harmonize circular economy principles with deep

decarbonization of the sector. Overall, our analysis reveals that it is technically possible to achieve net-zero emissions across the cement and concrete cycle by 2050 through the rapid and ambitious implementation of all the supply- and demand-side strategies spanning the entire value chain.

It is important to note that net-zero emissions, as defined here, refer to a state of equal CO$_2$ emissions and uptake, both of which change depending on strategy implementation (Fig. 4). With supply-side strategies alone, CO$_2$ emissions in 2050 (11 Mt-CO$_2$) far exceed CO$_2$ uptake and storage by natural carbonation and CCUs (−7 Mt-CO$_2$), resulting in net positive CO$_2$ fluxes in 2050 (4 Mt-CO$_2$). With both supply- and demand-side strategies in place, 2050 CO$_2$ emissions could be reduced more deeply (4 Mt-CO$_2$). Thus, the residual emissions are fully offset by CO$_2$ uptake and storage (−5 Mt-CO$_2$), providing net negative CO$_2$ fluxes (−1 Mt-CO$_2$) across the entire cement and concrete cycle. This condition refers to a state in which emissions associated with production activities are balanced by system-wide uptake and storage, mainly from in-use stock and demolition waste brought about by past cement and concrete production.

**Resource use implications**. Achieving net-zero emissions through both supply- and demand-side strategies substantially changes future trends in resource use (Fig. 5). In cases where both supply- and demand-side strategies are fully implemented, the per capita domestic consumption and in-use stocks of concrete are reduced by ~55% and 20%, respectively, by 2050 compared to 2020 levels. This, combined with a shrinking population, reduces total domestic consumption and in-use stocks of concrete by 62% and 33%, respectively, by 2050, compared to 2020 levels. While the importance of absolute decoupling of material consumption from economic growth is widely recognized[33] these findings imply the need for absolute decoupling of in-use stocks as well, rather than the relative decoupling due to stock saturation phenomenon observed in several high-income countries (Supplementary Fig. 24).

Such radical changes in the cement and concrete cycle have the co-benefit of reducing the use of scarce resources such as aggregates and water. Compared to the baseline scenario, fully implementing all of the supply- and demand-side strategies reduces virgin aggregate demand by 88% and batching water demand by 56% in 2050, and cumulatively by 54% and 38%, respectively, during 2020–2050 (Supplementary Fig. 28). These trends demonstrate the apparent synergistic effects of achieving

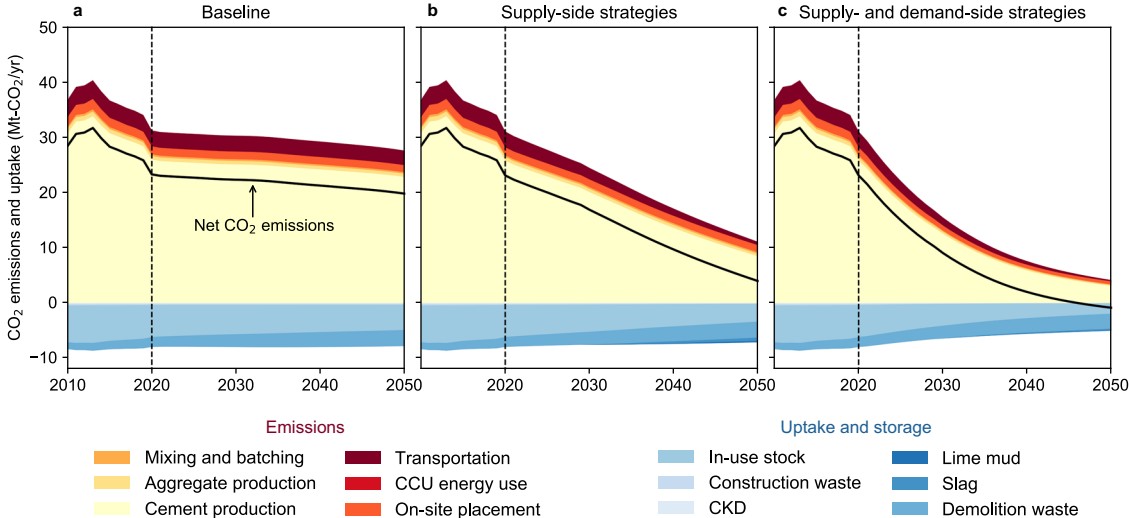

**Fig. 4 CO₂ emission and uptake associated with cement and concrete cycle in Japan under the three representative scenarios, 2010–2050. a** Baseline.
**b** Supple-side strategies. **c** Supple- and demand-side strategies. The vertical dashed lines mark the year in which the future scenarios begin (2020). CCU
carbon capture and utilization, CKD cement kiln dust.

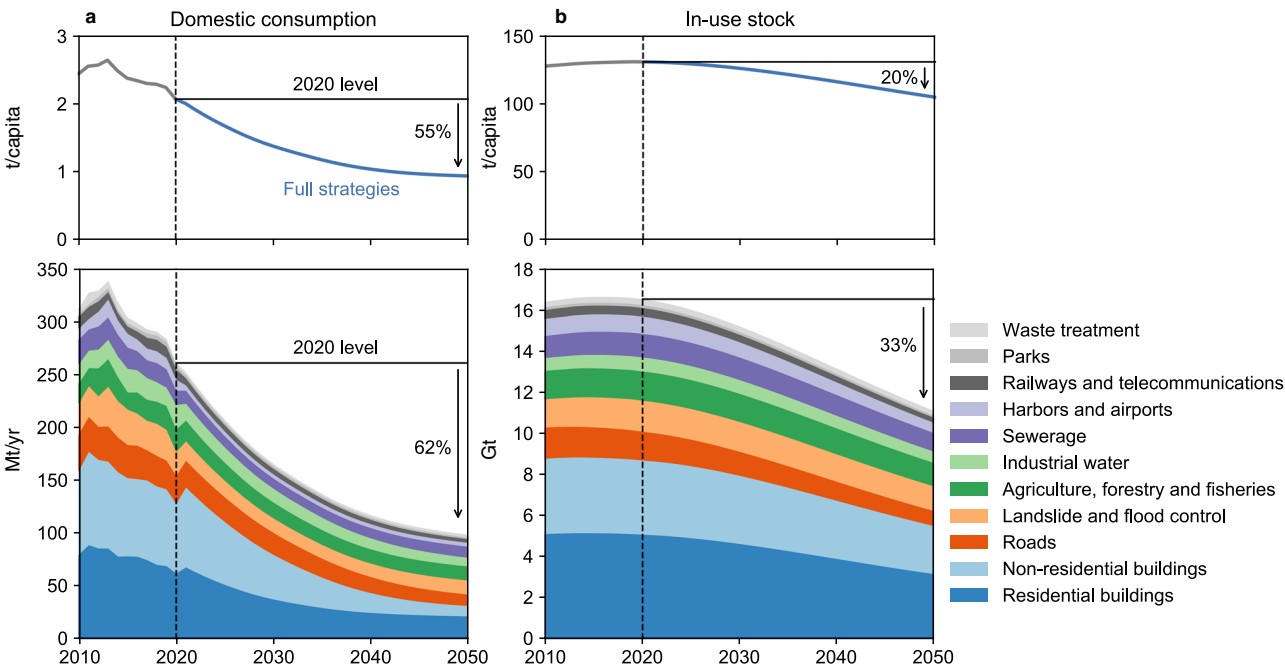

**Fig. 5 Domestic consumption and in-use stocks of concrete with and without a set of supply- and demand-side strategies in Japan, 2010–2050.**
**a** Domestic consumption. **b** In-use stock. The top panel shows per capita values, while the bottom panel shows the total values. The vertical dashed lines
mark the year in which the future scenario analyses begin (2020).

net-zero emissions in alleviating the sand crisis[34], air pollution[35], and water stress[36].

**The role of CCS**. Although CCS belongs to the lowest tier of mitigation strategies[21], it remains one of the central strategies in industry[19]. With this in mind, the role of CCS in achieving net-zero emissions is investigated by conducting a sensitivity analysis to better understand how demand-side strategies can reduce reliance on CCS (Fig. 6). If no demand-side strategy is implemented, then 70% of the kiln capacity will need to be equipped with CCS to achieve net-zero emissions by 2050. Such reliance on CCS can be reduced to 30% by implementing a demand-side strategy of just 50% of the maximum technical potential. When

the demand-side strategy is implemented at more than 80% of the technical potential, the need for CCS in achieving net-zero emissions by 2050 is completely eliminated. These results reflect the complementary relationship between CCS and demand-side strategies in decarbonizing this hard-to-abate sector.

**Discussion**
What this study clearly establishes is that, while net-zero emissions across the cement and concrete cycle are technically achievable, there is no "silver bullet" that can provide sufficient emission abatements on its own. Rather, a cross-cutting strategy that spans the entire cement and concrete cycle is required in order to achieve deep decarbonization in the limited time

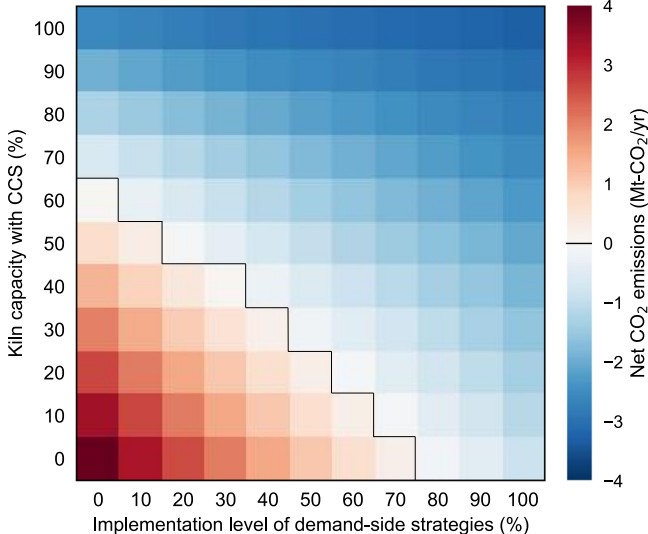

**Fig. 6 Sensitivity of 2050 net CO$_2$ emissions attributable to the cement and concrete cycle in Japan to the implementation level of demand-side strategies and the proportion of kilns equipped with carbon capture and storage (CCS).** The black lines mark the divisions between pathways that do (blue) and do not (red) achieve net-zero emissions. In all scenarios, all supply-side strategies are expected to be implemented except CCS, which is shown on the vertical axis.

remaining. This finding sheds new light on the decarbonization roadmap currently being pursued by governments and industry[9,19]. Although the Japanese government and industrial sector are pinning high hopes on supply-side strategies, including the use of alternative fuels and CCU, our analysis indicates that these alone will not likely bring about sufficient emission reductions in time without mass deployment of CCS. This is because conventional supply-side strategies already leave little room for improvement due to historical industry efforts, and emerging supply-side strategies may be limited in their potential by applicable concrete strength classes and the availability of industrial waste. That is, decarbonization through solo efforts by the cement and concrete industry will inevitably have to rely on CCS, which requires considerable time to deploy on a large scale and carries the risk of not being ready in time[37,38]. Collectively, our analysis calls into question the decarbonization roadmaps currently being proposed by government and industry, and emphasizes the importance of demand-side strategies to enable efficient use of cement and concrete in the built environment, in conjunction with supply-side strategies.

The challenge we ultimately face is how to promote demand-side strategies. Implementation of demand-side strategies requires more than just the efforts of the cement and concrete industry acting alone; it requires the concerted action of multiple stakeholders, including architects, urban planners, landowners, constructors, general consumers, and waste processors[6]. In this context, our work can strengthen the scientific basis of numerical targets for material flow-related indicators that provide a unified direction for the various stakeholders. The current Japanese national material flow targets were set before the declaration of carbon neutrality and do not necessarily guarantee consistency with net-zero emissions[39]. Our analysis bridges this gap by providing a set of indicators, i.e., domestic consumption and in-use stock, that are fully consistent with the net-zero emissions target. As demonstrated in the sensitivity analysis, since demand-side strategies and CCS can be complementary, placing the material flow-related indicators presented in this work as national targets will allow us to prepare for the risk of failure to scale up CCS sufficiently in time.

It is important to note that our analysis does not negate the need for investment in CCS. Rather, this study emphasizes that the arsenal required to decarbonize the cement and concrete sector should include strategic options that go beyond just "silver bullet" supply-side technologies. In particular, from a strategy standpoint, material-efficient design and more intensive use need to be a priority in light of their high emissions-saving potential. From the perspective of consumption activities, the key intervention domain includes housing, medical care, health care, transportation, education, and other services such as retail. These findings provide a window of opportunity for effective emission mitigation through targeted measures that enable and encourage consumers to seek out and adapt to material-efficient buildings and infrastructure in their consumption-related activities.

Despite the importance of demand-side strategies, there is absolutely no incentive for the materials industry to promote efficient material use, since the current profits are directly related to the volume of materials sold. More importantly, purchases of cement and concrete currently account for less than 10% of total spending in the building and infrastructure construction sector, suggesting the existence of weak economic incentives to pursue material efficiency, even on the demand side (Supplementary Figs. 31–33). These facts suggest that material efficiency gains will not progress spontaneously, but will require clear policy guidance. Since demand-side strategies, especially those involving material-efficient design and more intensive use, have been rarely covered by existing policies, adjustments will be needed to raise awareness of the importance of material efficiency, which is much less recognized than energy efficiency[40]. For example, as a condition for tax incentives, the "Low Carbon Building Certification Scheme"[41] only certifies buildings that are wooden, long-lived, or made of cement containing blast furnace slag or fly ash. Adding standards for cement and concrete usage, along with adjustments to building codes, could be an effective way to encourage material-efficient design. Another possibility is to amend the "Building Energy Efficiency Act"[42], which regulates compliance with energy efficiency standards, and the "Low Carbon City Development Plan"[41], which provides subsidies to selected municipalities. Since the current focus of these laws is primarily on operational energy use, articulating "embodied" carbon per specific functional unit (e.g., unit floor area) can add compelling reasons for stakeholders to build materially efficient buildings and urban structures. The newly enacted subsidy program for life-cycle carbon-minus houses is innovative in that it includes embodied carbon in the assessment[43]. However, since only the use of blast furnace slag-containing cement and lifetime extension are explicitly mentioned in the material-related efforts, adjustments will need to be made to encourage more diversified demand-side strategies. The key here is to recognize that the amount of cement and concrete that can be produced and used in a net-zero future is not infinite, and that it is considerably less than current levels. This means that stock growth and turnover must be managed simultaneously so as not to induce excessive use of low or zero carbon materials[44].

Importantly, the decarbonization pathway prescribed here depends on how CO$_2$ uptake from concrete carbonation and CCUs is accounted for in the inventory. Currently, CO$_2$ uptake from either natural carbonation or CCUs is not accounted for in national inventories under the Paris Agreement[45]. If this situation persists, the residual CO$_2$ emissions will need to be reduced in other ways. Since this can clearly make a substantial difference in the design of strategies and investments, there is an urgent need to start and settle the debate on how to account for concrete-related CO$_2$ uptake in national inventories under the Paris Agreement. If this is not done, the industry's vision of achieving net-zero emissions will be inconsistent with that at the national

level under the Paris Agreement, which will in turn jeopardize the effectiveness of net-zero emissions in the context of specific climate goals.

Overall, the message of this study is clear: cross-cutting strategies involving both the supply and demand sides can decarbonize the entire Japanese cement and concrete cycle by 2050 without resorting to mass deployment of CCS. However, realizing these strategies will require (1) coordinated policies to raise awareness of the importance of demand-side strategies, (2) revised material flow-related targets that are consistent with national commitments to climate change, and (3) consensus in the accounting methods used to assess $CO_2$ uptake by carbonation and CCUs in national inventories. These perspectives are not unique to Japan, given that most high-income countries in the world trace similar patterns of cement and concrete use. In particular, many middle- and low-income countries, which are in the process of expanding their material stocks[46,47], now have an important opportunity to systematically converge their stock growth at a much lower level than existing developed countries by incorporating efficient use of cement and concrete into urban development planning.

## Methods

**Mapping the cement and concrete cycle**. The analysis begins with a comprehensive understanding of the historical cement and concrete cycle, which is poorly understood due to fragmented statistical data. This challenge is addressed using a systems model encompassing a series of life-cycle stages, including raw material extraction, processing, manufacturing, in-use stocks, demolition, and recycling. The model first quantifies the flow and stock of concrete, based on construction activities and structure-specific material intensities, and then relates them to other material uses such as cement, clinker, aggregates, and water. The basis of the model is the law of mass conservation, which describes the structure and dynamic changes of material flows and stocks over time and space by a series of mass balance equations. The in-use stock of concrete is estimated using the inflow-driven dynamic MFA[48], assuming the specific lifetime of each end-use category. More specifically, this is a time-cohort-type approach that derives the in-use concrete stocks from the sum of the concrete inflows embedded in surviving buildings and infrastructure each year. Thus, the in-use stock of concrete is defined as the apparent quantity of material in buildings and infrastructure that are in-use in any given year. Note that since the focus of this study is on domestic demand, international trade is not taken into account here.

This study considers 11 end-use categories of concrete based on national official statistics: (i) residential buildings; (ii) non-residential buildings; (iii) roads; (iv) landslide and flood control; (v) agriculture, forestry and fisheries; (vi) industrial water; (vii) sewerage; (viii) harbors and airports; (x) railways and telecommunications; (xi) parks; and (xii) waste treatment. In this study, residential and non-residential buildings are further broken down into six sub-categories according to structure type: wood, steel-reinforced concrete, reinforced concrete, steel frame, concrete block, and others. System variables and parameters are determined by aggregating various fragmented statistical data on cement, readymixed concrete, aggregates, building, infrastructure, and material intensities. Some data are supplemented with information from previous MFA studies and interviews with industry representatives. A more detailed description of the modeling procedures and data sources is provided in Section 1.1 in the Supplementary Information.

**Calculating $CO_2$ emissions and uptake**. The $CO_2$ emissions associated with the cement and concrete cycle are calculated based on a comprehensive dataset documenting energy consumption and emission factors used in each process. This approach allows us to track the $CO_2$ emissions associated with electrical and thermal energy consumption, and chemical reactions (the conversion $CaCO_3 \rightarrow CaO + CO_2$ in the kiln) over the entire concrete cycle. The emission sources considered in this study are roughly divided into six major categories: cement production, virgin aggregate production, recycled aggregate production, concrete mixing and batching, concrete on-site placement, and transportation activities. Emissions from the use and dismantling phases are excluded from the model as it is difficult to assign them to a single material. A Japan-specific dataset on the energy consumed and the emissions generated in each process is compiled based on various statistical data and national emission inventories (see Supplementary Tables 12–14 and Figs. 6–10).

In addition to $CO_2$ emissions, $CO_2$ uptake from concrete carbonation is estimated using a physicochemical model[49–51]. Concrete carbonation is a phenomenon in which $CO_2$ diffuses into cementitious materials and reacts with hydrates, resulting in the gradual loss of alkalinity in cementitious materials. Although this phenomenon has traditionally been recognized as a durability issue

of reinforced concrete, its role as a $CO_2$ sink has gained attention in recent years in the context of climate change mitigation[4,29,49,51]. This study estimates the $CO_2$ uptake by concrete carbonation considering four sinks: concrete, mortar, construction waste, and cement kiln dust. The progress of carbonization is explicitly modeled based on Fick's law of diffusion. In this case, the carbonation rate of concrete and mortar is adjusted to account for the effects of exposed surface area, thickness, compressive strength class, exposure conditions, additives, atmospheric $CO_2$ concentration, coating and covering, and exposure time. The carbonation rates of demolished concrete and mortar are modeled assuming that the waste particles are spherical. Uncertainties in $CO_2$ uptake estimates are evaluated by Monte Carlo simulations in which each parameter is randomly extracted from a specific probability distribution. Details of the physicochemical model can be found in Section 1.3 of the Supplementary Information.

**Linking material flows with final demand drivers**. The process-based MFA described above only captures the material flows for each broad application category, such as buildings, and does not provide any insights into what final demand actually drives them. For example, non-residential buildings can be used as schools or hospitals to meet needs for education or medical services. Thus, to better understand the demand-side drivers of the current cement and concrete cycle, we link estimated material flow data to final demand drivers based on the input-output approach. Specifically, cement flows in 2019 are linked to ~400 sectors using the 2015 Japanese input-output table, which is the most recent data currently available[52]. In this case, the Japanese input-output table currently treats fixed capital formation (e.g., infrastructure, machinery, and transport equipment) as a final demand sector rather than as inputs to the production system[53,54]. Consequently, the typical input-output analysis does not tell us which final demand is really driving the use of construction materials, including cement and concrete, which are used mainly in the production of fixed capital stocks. We, therefore, endogenize fixed capital by using an augmentation method[55] with the fixed capital formation table in the supplementary material of the Japanese input-output table. Please see Section 1.4 and Supplementary Table 29 in the Supplementary Information for details on calculations and consumption categories in the input-output table.

**Developing a baseline scenario**. The future concrete flows and stocks are explored and tied with building and infrastructure stock dynamics. In this approach, future per capita building floor space and infrastructure stock are first estimated based on historical observations. A stock-driven dynamic MFA then translates them into newly constructed buildings and infrastructure[48]. Ultimately, using material intensity data for each application, dynamic changes in the flow and stock of buildings and infrastructure can be tied to a series of concrete and associated resource uses.

We define a baseline scenario as a future in which the way that buildings and infrastructure are manufactured, used, and demolished remains unchanged. This scenario is based on the observation that per capita building floor space and infrastructure stock have been stable since around 2010 (Supplementary Figs. 22, 23) and assumes that these values will remain constant into the future. Such an assumption reflects the fact that the buildings and infrastructure that are required to maintain our high standard of living have already been sufficiently established in Japanese society. The baseline scenario is interpreted as a benchmark for understanding the $CO_2$ emission mitigation potential of additional strategies. The future population is based on the Shared Socioeconomic Pathway 2, representing a "middle-of-the-road" pathway with moderate population and GDP growth[56].

**Exploring strategies for achieving net-zero emissions**. We consider a total of 16 strategies to achieve net-zero emissions in the cement and concrete sector. These can be broadly classified into three categories: conventional supply-side strategies (six strategies), emerging supply-side strategies (three strategies), and demand-side strategies (seven strategies). A brief description of each strategy group is given below, with detailed technical descriptions, assumed parameters, and barriers to implementation to be found in Section 3 of the Supplementary Information.

*Conventional supply-side strategies.* Strategies in this category are envisioned to enhance current efforts by industries to reduce $CO_2$ emissions. Such measures include improving thermal and electrical efficiency, using low-carbon fuels, reducing clinker-to-cement ratios, lowering transportation emissions, and decarbonizing the electricity supply. The technological potential of these strategies is determined based on best practices from around the world and the long-term vision of the industry to reflect an ambitious, but not unfeasible level of implementation. Although the speed of diffusion of each strategy is influenced by multiple factors, including political and regulatory procedures, infrastructure development, and information accumulation, for simplicity, the speed of implementation is modeled assuming linear growth from 2021 to 2050[37].

*Emerging supply-side strategies.* This category includes innovative supply-side interventions, but could take a little time to scale up: using low-carbon cement chemistries and CCUs. We consider six low-carbon cement chemistries that have been identified as being commercially viable in the next decade or so[6,7] reactive belite cement, belite-ye'elimite-ferrite cement, carbonatable calcium silicate cement,

calcium sulfoaluminate, Celitement, and magnesium oxides derived from magnesium silicates. As for CCU, there are two types of CCU to be considered: "concrete curing", where $CO_2$ gas is injected during the batching and mixing of concrete or during the curing process of precast products, and "mineralization", where $CO_2$ is mineralized with alkaline substances such as calcium and magnesium to form carbonate minerals. CCU technologies mitigate $CO_2$ emissions through increased $CO_2$ uptake and storage over the service life of the concrete and reduced binder due to an increase in the compressive strength of concrete. The technical potential of these strategies is established based on the applicable strength classes and the availability of industrial wastes. Based on the government's roadmap[57] and a detailed technical review[7], the speed of implementation is assumed to be through linear growth from 2030 to 2050.

*Demand-side strategies (material efficiency strategies).* Material efficiency strategies refer to demand-side interventions by systems and actors that use concrete in the built environment, such as architects, urban planners, property owners, constructors, general consumers, and waste processors. A total of seven interventions across the cement and concrete cycle are considered here: material-efficient design, construction waste reduction, more intensive use, lifetime extension, component reuse, downcycling, and stockpiling of demolition waste. Since our focus is on the cement and concrete cycle, we do not consider material substitution with, for example, engineered wood[58,59]. Modeling of its effects requires going beyond the study of a single material and tracing a set of materials together, for example, building materials[24]. The technical potential of each strategy is set based on various scientific studies. Since strategies in this category can be implemented immediately[23], implementation is envisioned to begin in 2021, as is the case with conventional supply-side strategies.

It is important to note that the demand-side strategies considered here are not independent and that they influence each other. A prominent example is the strategies related to lifetime extension and several end-of-life options: the longer a concrete structure lasts through lifetime extension, the less end-of-life material is available for component reuse, downcycling, or waste stockpiling. Our model captures such interactions through a set of mass balance equations that ensure the feasibility of each strategy in terms of mass balance constraints. The interaction of strategies that do not depend on mass balance constraints is further explained in Section 3 of the Supplementary Information.

**Quantifying the role of CCS in relation to demand-side strategies.** While this study sets out to explore a net-zero emission scenario that does not rely on CCS, the role of CCS is investigated in the manner of a sensitivity analysis. Doing so allows us to better understand how demand-side strategies can reduce reliance on CCS. Specifically, we quantify the extent to which different implementation levels of demand-side strategies reduce dependence on CCS. This is done by estimating the net $CO_2$ emissions when the implementation level of demand-side strategies and the percentage of kiln capacity equipped with CCS are varied from 0% to 100% in 10% intervals. Thus, a total of 121 different scenarios are investigated. In this case, three possible types of CCS can be used in kilns: oxy-fuel firing, pre-combustion, and post-combustion, where $CO_2$ is separated and captured during combustion, pre-combustion, and post-combustion, respectively. Here, the implementation of post-combustion with chemical absorption is determined by referring to the literature[60], taking into account its capture efficiency (90%) and energy penalty (2070 MJ/t-$CO_2$ captured)[6,61].

## Data availability
The input data and model results of this study have been deposited on GitHub (https://github.com/takumawatari/concrete_cycle_jp). Permanent references to the data are also accessible through the Zenodo repository[62]. Source data are provided with this paper.

## Code availability
The full model code is available on GitHub (https://github.com/takumawatari/concrete_cycle_jp). Permanent references to the code are also accessible through the Zenodo repository[62].

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

## Acknowledgements

This research was supported in part by a Grant-in-Aid for Research (No. 21K12344) from the Japanese Ministry of Education, Culture, Sports, Science and Technology, and by the Environment Research and Technology Development Fund (JPMEERF20S20604 and JPMEERF20223001).

## Author contributions

T.W. and C.Z. designed the study; T.W., C.Z., and S.H. performed the analyses; and T.W., C.Z., S.H., and K.N. prepared the manuscript.

## Competing interests

The authors declare no competing interests.
