## [Peer Review File · Nature Communications]

Response to reviewers, first round review –

Reviewer #1 (Remarks to the Author):

The authors present an important study looking at the importance of supply-side interventions in mitigating emissions in construction.

The introduction is well constructed, and rightly focuses on the scientific literature. Because of the interest and importance of the topic, there is much grey literature looking at the decarbonation of cement, notably the GCCA's roadmap. It would be useful to contrast scientific literature and industry pledges.

I 100 How did the authors obtain the figure of the amount of carbon uptake? It looks like it is coming from their own model, but then a range would give higher confidence.

I. 128 a comparison to international trends would be good and would solidify the author's claim.

I. 144 it is not clear how the authors know what is the maximal theoretical efficiency of a kiln. In effect, a modern kiln uses about twice the amount of energy thermodynamics require for decarbonation and clinkering, but there is some variation, and suboptimal control of operating conditions can cause emissions to be up to 12% higher than achievable.

I. 148 It is not clear that these SCMs will be available in the future: BOS furnaces and coal plants are not likely part of a 1.5C degree future...

I 174 It is unclear how stockpiling leads to faster carbonation over landfilling

I 202 and next. I agree with the authors, but more crucially, CCS works well with centralised production but not so much for things like cement which are highly local.

In general, the authors present an important contribution, which shows that there are alternatives to CCS as a catch-all which will reduce our emissions to zero.

An aspect that the authors could detail some more is that some strategies may not be independent: for example using recycled aggregates which are carbonated depends on a certain demolition rate, which is not compatible with the extension of structures. Similarly, component reuse. This should be reviewed.

The authors should check whether the design strategies described elsewhere in the literature are also applicable to Japanese practice. Other more recent work suggest that there are fundamental aspects to design which coincidentally might lead to a very similar number.

The effect is quite small in any case, but it is unreasonable to assume no waste in construction. Further, much waste in construction, for which it is difficult to get number may be in discarded wrappings, plasterboard off-cuts, etc.

Figure S4 is probably wrong.

Figure S7 deserves comments: in most of the world the mix is 50:50 coke and petcoke. Maybe this is worth making the distinction as these fuels have different calorific power.

In general, this is a good paper making a very important point. It would be good if some of the modelling aspects were verified, crucially how independent the strategies can be. There would be value in discussing how applicable the results are outside of Japan: some of the conclusions are likely applicable generally to ageing, developed countries, but some aspects may be more broadly applicable.

Reviewer #2 (Remarks to the Author):

The paper review known technologies and apply that on US cement emissions.
The paper is extremely clear and makes a useful distinction between supply and demand measures for cement.
I think issues are well covered.

On the demand measures, I would have included the fact that we could provide same function as concrete products with other type of materials.
This can be discussed.
Wood would be the easiest material as replacement.
Maybe referring the classic study on "building as carbon sinks"
Churkina et al. 2020. Buildings as a global carbon sink. Nature Sustainability.
But also pointing the limit of such substitution in term of material availability.
Pomponi et al. 2020. Buildings as a Global Carbon Sink? A Reality Check on Feasibility Limits. One earth

second material with much more availability and similarity in term of concrete use would be excavation materials.
it's abundant and can be used as poured earth technology.

I would also have questioned the link between m²/cap and GDP.
Increase in GDP is linked with increase in m²/cap (and so concrete demand). Is it possible to decouple growth from m² consumption?

Reviewer #3 (Remarks to the Author):

The paper is well written and interesting. The study presents potential pathways to decarbonize the Japanese concrete sector combining both supply-side and demand-side interventions. Unlike the mainstream narrative, the authors argue that to reach carbon neutrality we must rely also on demand-side strategies. Results are noteworthy, the work is well-presented and fairly innovative, but the authors should provide additional evidence and explanations for the manuscript to be considered for publication.

- Carbon emissions accounting. My main concern regards the way emissions were calculated for the decarbonisation pathways. If CCS is not included, the only way I see that the sector becomes carbon neutral is that all of the fossil CO₂ emitted in the production phase is reabsorbed by the concrete during its lifetime. Reducing the demand for concrete does not lead to carbon neutrality if the CO₂ emitted to produce that concrete is not reabsorbed or stored. Please elaborate.

- Study motivation. The authors present decarbonisation pathways without CCS "due to concerns about its social acceptability, speed of diffusion, and technology lock-in". However, no information is provided in the manuscript regarding the social acceptability of the alternative strategies proposed.

- Time horizon. Although 2050 was chosen to align with the international climate targets, are there insurmountable barriers that would prevent the Japanese concrete sector to reach carbon neutrality before 2050? If the authors were in control of the policies, how fast would they decarbonize the sector?

- Supply-side strategies

- o Low-carbon fuel utilization. Why the use of biomass and hydrogen was not considered? How were the emissions from natural gas (including leakage) and waste fuels modelled? What would be the fate of waste if not used for cement? Please report the main emission factors used for the

study in the text.

o Clinker-to-cement ratio reduction.

- ♣ Why only a 40% replacement was considered? For instance, the use of calcined clay limestone cements could already achieve replacements of 50% [1].
- ♣ How was the clinker to SCM substitution ratio modelled? 1-to-1? The ratio changes depending on the strength and exposure conditions of concrete [2].
- ♣ From figure S17 there seems to be an abundance of slag compared to the use. However, from the reference reported ([48]), it looks like all the slag produced is already used. Why the discrepancy between the source and figure S17 (e.g., for 2017: total produced 23.03 kt, used 23.97)? Moreover, the amount of slag will probably reduce in the future thanks to fuel/technology changes in steel production [3]. How does this affect the results?

o CCU. The role of CCU seems to play a crucial role in the abstract. However, not enough information is provided in the text regarding the technology and the emission accounting.

- ♣ Does CCU refer only to concrete curing and mineralization? Is the CO₂ captured from the kiln?
- ♣ How does CCU affect carbonation? I.e., CO₂-cured concrete will absorb the same amount of CO₂ in its lifetime than traditional concrete?
- ♣ Why CCU for non-concrete applications (e.g., fuel and chemical production) was not included?
- ♣ Line 266: What does it mean that CCU is excluded from national inventories? Emissions are accounted for the CO₂ user but not for the CO₂ producer? Please provide a reference and explain.
- ♣ Although I agree there is an urgent need for a clear methodology, it is clear what carbon neutral means: the concentration of CO₂ (and other greenhouse gases) in the atmosphere should not increase due to cement and concrete production.

• Demand-side strategies. More information should be provided regarding the implementation of these strategies.

o What are the barriers for each strategy?

o Why a gradual implementation curve was considered for more intensive use? I.e., what is the issue with abrupt changes?

o How is the lifetime extension modelled? How does refurbishment/maintenance affect the greenhouse gas balance? How is it ensured a longer lifetime?

o How is the efficient design strategy implementation modelled?

• Carbonation.

o Is concrete assumed to be pulverized at its end of life to optimize the carbonation? If yes, how does the energy consumption affect the CO₂ balance? If concrete was landfilled, how much CO₂ would be absorbed?

o Line 100: How was the CO₂ uptake in 2019 calculated? Is it just from in-use stock or from waste concrete?

o Line 340: please specify the range of carbonation assumed for the different concrete types: e.g., 10%-15% of the calcination emissions are reabsorbed during the service life of concrete, additional 10-15% at the end of life.

• Applicability.

o What policies should be brought forward to reach carbon neutrality according to the authors? From the work, it looks like more intensive use and efficient design are the most efficient options. How can they be implemented?

o Line 248: please specify the material flow-related indicators that you would place as national targets

Other questions:

• Line 123: Does that mean that the amount of waste concrete equals the amount of new concrete?

• Line 190: How could the resource use decoupling be implemented in countries that did not reach stock saturation?

• Line 304: What's the difference between steel-reinforced concrete and reinforced concrete?

• Fig 4: Is carbonation included in the figure? How?

• Supp. 5. Why cementitious solidifiers were not included in the assessment if they represent a

significant share of cement use?

Typos and small edits:

- Line 36: I disagree that “the question regarding what actions need to be taken to decarbonize the cement sector is largely unexplored”, as it is proven by the following paragraph.
- Line 62: avoid the use of non-scientific terms as “huge”
- Line 206: after better a verb is missing

[1] K. Scrivener, F. Martirena, S. Bishnoi, S. Maity, Calcined clay limestone cements (LC3), *Cem. Concr. Res.* 114 (2018) 49–56. <https://doi.org/10.1016/j.cemconres.2017.08.017>.

[2] A. Arrigoni, D.K. Panesar, M. Duhamel, T. Opher, S. Saxe, I.D. Posen, H.L. MacLean, Life cycle greenhouse gas emissions of concrete containing supplementary cementitious materials: cut-off vs. substitution, *J. Clean. Prod.* 263 (2020) 121465.

<https://doi.org/https://doi.org/10.1016/j.jclepro.2020.121465>.

[3] International Energy Agency, Cement Sustainability Initiative, Technology Roadmap. Low-Carbon Transition in the Cement Industry, Paris, France, 2018.

https://doi.org/10.1007/SpringerReference_7300.

Response to reviewers

Reviewer #1 (Remarks to the Author):

Comment

The authors present an important study looking at the importance of supply-side interventions in mitigating emissions in construction.

>Response

Thank you very much for your valuable time spent reading our paper and for your insightful feedback. We have attempted to address all of your concerns. Please see our point-by-point responses and the revisions that we have made to the manuscript.

Comment

The introduction is well constructed, and rightly focuses on the scientific literature. Because of the interest and importance of the topic, there is much grey literature looking at the decarbonation of cement, notably the GCCA's roadmap. It would be useful to contrast scientific literature and industry pledges.

>Response

We thank the reviewer for this insightful comment. We have added the following text and references to demonstrate the reliance on CCS in the industry pledges.

“Among these various supply-side measures, industry pledges and policy discussions are particularly dependent on CCS¹⁸⁻²⁰, even though it belongs to the lowest hierarchy of strategies due to technology lock-in concerns and low resource efficiency²¹.” (line 43)

Comment

I 100 How did the authors obtain the figure of the amount of carbon uptake? It looks like it is coming from their own model, but then a range would give higher confidence.

>Response

We apologize for our unclear explanation of the carbon uptake calculations in the previous manuscript. The carbon uptake was calculated using a physicochemical model. We have revised the text and figure to better clarify this point.

“The CO₂ uptake by concrete carbonation reached 8.2 Mt-CO₂ (7.1-10.0 Mt-CO₂ interquartile range) in 2019, equivalent to approximately 24% of concrete-related emissions and approximately 1% of Japan’s total emissions. In-use stocks act as the largest sink during the service life of buildings and/or infrastructure, absorbing approximately 74% of the total uptake. Another 22% is derived from demolition waste at the end-of-life stage, and the remaining 4% from cement kiln dust and construction waste (Fig. S26 in the Supplementary Information).” (line 101)

Fig. 1 Cement and concrete cycle and associated CO₂ fluxes in Japan in 2019. All flows are shown to scale in Mt/year; in-use stocks shown in the box scaled differently than flows. The numbers in the lower circle show annual CO₂ emissions and uptake associated with the concrete cycle in Mt-CO₂/year. SCMs: supplementary cementitious materials. CKD: cement kiln dust. The Sankey diagram was designed with floWeaver⁶⁸.

Fig. S26 CO₂ uptake from concrete carbonation, 1950-2019.

Comment

I. 128 a comparison to international trends would be good and would solidify the author's claim.

>Response

Thank you for this insightful comment. We have added a figure and text to compare cement stocks in several developed countries.

“Such a trend is not unique to Japan, but is common to many high-income countries (Fig. S24 in the Supplementary Information).” (line 134)

Fig. S24 International comparison of in-use stock of cement equivalent, 1950-2020. Data for countries other than Japan are based on our previous study [15], and country selection is based on other material flow analysis studies [65].

Comment

I. 144 it is not clear how the authors know what is the maximal theoretical efficiency of a kiln. In effect, a modern kiln uses about twice the amount of energy thermodynamics require for decarbonation and clinkering, but there is some variation, and suboptimal control of operating conditions can cause emissions to be up to 12% higher than achievable.

>Response

Thank you for this comment. The explanation of this section was unclear and we have modified the text and figure to better clarify this point. Please note that we adopted 'average thermal efficiency of the global 10% best in class' as our target, and not 'maximum theoretical efficiency'.

“Furthermore, reflecting the fact that the thermal and electrical energy efficiency of Japanese cement facilities is already close to the global 10% best-in-class, energy efficiency improvements would mitigate emissions by only 3% in 2050.” (line 148)

Fig. S13 Thermal efficiency in the cement kiln, 2006-2050.

Comment

I. 148 It is not clear that these SCMs will be available in the future: BOS furnaces and coal plants are not likely part of a 1.5C degree future...

>Response

We agree with the reviewer and have revised our assumptions so that these two SCMs are not used as clinker substitutes.

“Since most of the CO₂ emitted during the cement production process comes from clinker production, reducing the clinker-to-cement ratio can contribute to emission savings. Clinker can be replaced by a variety of supplementary cementitious materials (SCMs), such as fly ash, ground granulated blast furnace slag (GGBFS), and calcined clay [35]. In Japan, GGBFS and fly ash are mainly standardized and used. However, the availability of these industrial by-products depends on steel production and coal-fired power generation, and their availability is expected to decrease in a net-zero future. Therefore, we focus on calcined clays, especially in combination with limestone (i.e., LC³ technology). Since the LC³-50 blend (50% clinker, 30% calcined clay, 15% limestone and 5% gypsum) has already proven to be capable of providing mechanical properties comparable to standard ordinary Portland cement as well as provide some durability improvements [36], this study adopts an aggressive target of 50% clinker replacement by 2050 [37]. Although the substitution ratio of SCMs to clinker can vary with compressive strength, workability, or exposure conditions, a one-to-one substitution ratio is assumed here, as in several existing studies [24,38].” (SI)

Comment

I 174 I ti unclear how stockpiling leads to faster carbonation over landfilling

**>Response**

The reason why stockpiling leads to faster carbonation than landfilling is because stockpiling demolition waste increases the surface area of crushed concrete exposed to air and allows more time for accelerated carbonation compared to a landfill environment. To better clarify this point we have modified our description of the waste stockpiling strategy as follows:

"During demolition, concrete is crushed into small pieces, which increases the surface area and promotes the carbonization process. Crushed concrete pieces are stored in intermediate treatment facilities for an average of 0.2 years in Japan [33]. Extending this storage period will promote the $CO_2$ absorption process of demolished concrete since the surface area of crushed concrete exposed to air is limited in a landfill environment [61]. This study assumes that the stockpiling period of demolition waste can be extended to one year by 2050 [24]." (SI)

**Comment**

l 202 and next. I agree with the authors, but more crucially, CCS works well with centralised production but not so much for things like cement which are highly local.

**>Response**

This is indeed a good point. We have added a table summarizing the barriers to achieving each strategy, including CCS.

**Table S28** Barriers in the implementation of several supply- and demand-side strategies. Based on a critical review of 37 previous studies [64], those mentioned more than once are checked.

| | | Energy efficiency improvement | Low-carbon fuel utilization | Clinker-to-cement ratio reduction | Low-carbon cement chemistries | CCUS | Material efficiency improvement |
|---|---|---|---|---|---|---|---|
| Economy | Higher cost | ✓ | ✓ | ✓ | ✓ | ✓ | ✓ |
| | Availability of materials | | ✓ | ✓ | ✓ | | |
| | Market uncertainty | | | | | ✓ | |
| | Lack of demand | ✓ | | ✓ | ✓ | ✓ | ✓ |
| | Market acceptance | | | ✓ | ✓ | | |
| | Fragmented supply chain | | | | | | ✓ |
| Technical | Lack of infrastructure | | ✓ | | | ✓ | |
| | Lack of expertise | ✓ | | | | | |
| | Poor information | ✓ | | | | | ✓ |
| | Additional energy requirements | | | | | ✓ | |
| | Time constraints | | | | | | ✓ |

	Longer curing time		✓	✓	
	Structural strength		✓		
Regulatory	Regulatory requirements	✓			✓
	Complex bureaucracy	✓		✓	
	Risk concern				✓
	Lack of effective policies		✓	✓	✓
	Institutional challenges	✓	✓		
Social	Social acceptance	✓		✓	
	Industry culture				✓

Comment

In general, the authors present an important contribution, which shows that there are alternatives to CCS as a catch-all which will reduce our emissions to zero.

An aspect that the authors could detail some more is that some strategies may not be independent: for example using recycled aggregates which are carbonated depends on a certain demolition rate, which is not compatible with the extension of structures. Similarly, component reuse. This should be reviewed.

>Response

Thank you for highlighting this important point. Indeed, the interactions among each demand-side strategy are critically important. We have added the following discussion.

"3.2 Interaction of each demand-side strategy

It is important to note that the demand-side (material efficiency) strategies considered here are not independent and that they influence each other. A prominent example is the strategies related to lifetime extension and end-of-life options: the longer a concrete structure lasts through lifetime extension, the less demolition waste is available for reuse, downcycling, or waste stockpiling. Our model captures such interactions through a set of mass balance equations that ensure the feasibility of each strategy in terms of mass balance constraints.

However, there are also interactions among strategies that do not depend on mass balance constraints. For example, optimizing concrete components for material-efficient design may reduce the potential for component reuse, which requires standardization [58]. In addition, more intensive use may inhibit service life due to premature deterioration associated with high frequency of use of concrete structures [62]. Clearly, further research is needed on the relative quantitative influence of factors on the model parameters associated with such interactions among strategies.

In addition, interactions beyond the system boundary of the cement and

concrete cycle are also important. For example, building longevity will slow the spread of better insulated buildings unless there are initiatives to retrofit existing buildings [63]. On the other hand, reductions in per capita floor space demand through space sharing and teleworking practices have the synergistic effect of reducing energy use during the utilization phase [57]. Modeling the synergistic effects of such interactions is another important task that needs to be undertaken in the future.“ (SI)

Comment

The authors should check whether the design strategies described elsewhere in the literature are also applicable to Japanese practice. Other more recent work suggest that there are fundamental aspect to design which coincidentally might lead to a very similar number.

>Response

Thank you for this thought-provoking comment. We have reviewed our assumptions regarding material-efficient design and have revised the text as follows:

“Performance-based design allows architects and contractors to design concrete mixtures that meet the necessary mechanical and durability specifications with less cement. Some evidence suggests that the cement content of concrete could be reduced by 15-20% without compromising compressive strength [47,48]. Importantly, a previous study showed that prescriptive design, which specifies the allowable cementitious content in concrete, induces over-use of cement [49]. According to their estimates, using performance-based design rather than prescriptive design could reduce carbon emissions per required volume of concrete by up to 30%. Here, we assume that the cement content of concrete can be reduced by 15% by 2050 [38]. Furthermore, precast components allow designers to manufacture concrete components with greater precision and reliability using less cement. Post-tensioning techniques can make parts of concrete elements thinner by stressing the rebar in the concrete floor slab before applying external loads. Changes in the way elements are specified could reduce the amount of cementitious materials used in structural components, especially as overdesign is often employed by designers when constructing structural elements. Together, we assume that a 13% reduction in the concrete intensity of concrete structures can be achieved without interfering with its function by 2050 [24,38]. Currently, the concrete intensity of buildings in Japan is two to four times higher than in the U.S. and China. This difference is mainly due to the structure of the building, especially the thickness of the foundation, piles, and columns [50]. Since the strategies considered here do not compromise strength or durability of structures, we assume that transitions can be encouraged by making adjustments to building codes and through appropriate code enforcement.” (SI)

Fig. S17 International comparison of concrete intensity of newly constructed residential buildings. Data adapted from [24].

Fig. S18 International comparison of concrete intensity of newly constructed non-residential buildings. Data adapted from [24].

Comment

The effect is quite small in any case, but it is unreasonable to assume no waste in construction. Further, much waste in construction, for which it is difficult to get number may be in discarded wrappings, plasterboard off-cuts, etc.

>Response

Thank you for this insightful comment. First, we have tried to strengthen the validity of our assumed construction waste generation rate by adding Japan-specific evidence based on interviews and surveys. Second, to avoid unrealistic assumptions, we have changed our assumption to a 1% improvement, rather than complete avoidance of construction waste.

Comment

Figure S4 is probably wrong.

>Response

Thank you for bringing this to our attention. The figure captions have been revised.

Comment

Figure S7 deserves comments: in most of the world the mix is 50:50 coke and petcoke. Maybe this is worth making the distinction as these fuels have different calorific power.

>Response

We thank the reviewer for highlighting this point. We obtained these data from industrial sources that differentiated between coal and petroleum coke. These data, combined with individual emission intensity data, are now used throughout the paper.

Fig. S8 Fuel mix of thermal energy consumption, 1999-2020. Data adapted from [7].

Comment

In general, this is a good paper making a very important point. It would be good if some of the modelling aspects were verified, crucially how independent the strategies can be. There would be value in discussing how applicable the results are outside of Japan: some of the conclusions are likely applicable generally to ageing, developed countries, but some aspects may be more broadly applicable.

>Response

We thank the reviewer for this insightful comment. As indicated above, we have added a discussion on the independence of each strategy. Further, we have added text on how the results from this study apply to regions outside Japan:

“Overall, the message of this study is clear: cross-cutting strategies involving both the supply and demand sides can decarbonize the entire Japanese cement and concrete cycle by 2050 without resorting to mass deployment of CCS. However, realizing these strategies will require (1) coordinated policies to raise awareness of the importance of demand-side strategies, (2) revised material flow-related targets consistent with national commitments to climate change, and (3) consensus in the accounting methods used to assess CO₂ uptake by carbonation and CCUs in national inventories. These perspectives are not unique to Japan, given that most high-income countries in the world trace similar patterns of cement and concrete use. In particular, many middle- and low-income countries, which are in the process of expanding their material stocks^{49,50}, now have an important opportunity to systematically converge their stock growth at a much lower level than the current developed countries by incorporating efficient use of cement and concrete into urban development planning.” (line 323)

Reviewer #2 (Remarks to the Author):

Comment

The paper review known technologies and apply that on US cement emissions. The paper is extremely clear and makes a useful distinction between supply and demand measures for cement. I think issues are well covered.

>Response

Thank you very much for your valuable time spent reading our paper and for your insightful feedback. We have attempted to address all of your concerns. Please see our point-by-point responses and the revisions that we have made to the manuscript.

Comment

On the demand measures, I would have included the fact that we could provide same function as concrete products with other type of materials.

This can be discussed.

Wood would be the easiest material as replacement.

Maybe referring the classic study on "building as carbon sinks"

Churkina et al. 2020. Buildings as a global carbon sink. Nature Sustainability.

But also pointing the limit of such substitution in term of material availability.

Pomponi et al. 2020. Buildings as a Global Carbon Sink? A Reality Check on Feasibility Limits. One earth

second material with much more availability and similarity in term of concrete use would be excavation materials. it's abundant and can be used as poured earth technology.

>Response

We thank the reviewer for highlighting this important point. Indeed, the possibility of material substitution has been deeply considered by us as well. In order to properly assess the impact of material substitution, a set of materials needs to be added to the system boundary to avoid double counting and to check the supply-demand balance. Since such efforts extend beyond the scope of this study, which solely focuses on the cement and concrete cycles, material substitution is not considered here. To clarify this point, we have added following text:

"Since our focus is on the cement and concrete cycle, we do not consider material substitution with, for example, engineered wood^{64,65}. Modeling of its effects requires going beyond the study of a single material and tracing a set of materials together, for example, building materials²⁴." (line 488)

Comment

I would also have questioned the link between m2/cap and GDP. Increase in GDP is linked with increase in m2/cap (and so concrete demand). Is it possible to decouple growth from m2 consumption?

>Response

We thank the reviewer for this insightful comment. To address your question, we plotted the relationship between per capita newly constructed building floor area and GDP. We found that these two variables are already decoupled. However, building stock and GDP are not decoupled. A more detailed analysis of the entire economic structure beyond just building construction and concrete is therefore needed to fully answer your question, but we hope that the figures we have added will help to clarify this point.

Fig. S25 Relationship between per capita GDP and key variables related to cement and concrete cycle.

Reviewer #3 (Remarks to the Author):

Comment

The paper is well written and interesting. The study presents potential pathways to decarbonize the Japanese concrete sector combining both supply-side and demand-side interventions. Unlike the mainstream narrative, the authors argue that to reach carbon neutrality we must rely also on demand-side strategies. Results are noteworthy, the work is well-presented and fairly innovative, but the authors should provide additional evidence and explanations for the manuscript to be considered for publication.

>Response

Thank you very much for your valuable time spent reading our paper and for your insightful feedback. We have attempted to address all of your concerns. Please see our point-by-point responses and the revisions that we have made to the manuscript.

Comment

- Carbon emissions accounting. My main concern regards the way emissions were calculated for the decarbonisation pathways. If CCS is not included, the only way I see that the sector becomes carbon neutral is that all of the fossil CO₂ emitted in the production phase is reabsorbed by the concrete during its lifetime. Reducing the demand for concrete does not lead to carbon neutrality if the CO₂ emitted to produce that concrete is not reabsorbed or stored. Please elaborate.

>Response

We thank the reviewer for highlighting this important point and apologize for our unclear explanation in the previous manuscript. Indeed, the essential condition for the concrete sector to become carbon neutral without CCS is that CO₂ emissions from production activities should match CO₂ uptake through carbonation or CCU. Our model defines exactly this condition as net-zero emissions. To better clarify this point, we have added following text and figure.

“It is important to note that net-zero emissions, as defined here, refer to a state of equal CO₂ emissions and uptake, both of which change depending on strategy implementation (**Fig. 4**). With supply-side strategies alone, CO₂ emissions in 2050 (11 Mt-CO₂) far exceed CO₂ uptake and storage by natural carbonation and CCUs (-7 Mt-CO₂), resulting in net positive CO₂ fluxes in 2050 (4 Mt-CO₂). With both supply- and demand-side strategies in place, 2050 CO₂ emissions could be reduced more deeply (4 Mt-CO₂). Thus, the residual emissions are fully offset by CO₂ uptake and storage (-5 Mt-CO₂), providing net negative CO₂ fluxes (-1 Mt-CO₂) across the entire cement and concrete cycle.” (line 192)

Fig. 4 CO₂ emission and uptake associated with cement and concrete cycle in Japan under the three representative scenarios, 2010-2050. The vertical dashed lines mark the year in which the future scenarios begin (2020). “Uptake and storage” includes CO₂ absorption by carbonation and CCU.

Comment

- Study motivation. The authors present decarbonisation pathways without CCS “due to concerns about its social acceptability, speed of diffusion, and technology lock-in”. However, no information is provided in the manuscript regarding the social acceptability of the alternative strategies proposed.

>Response

This is a good point. Indeed, we have not been able to assess the social acceptability of alternative strategies. To clarify this point in the text, we emphasized that the selection of strategies is based simply on the strategy hierarchy:

“Among these various supply-side measures, industry pledges and policy discussions are particularly dependent on CCS^{18–20}, even though it belongs to the lowest hierarchy of strategies due to technology lock-in concerns and low resource efficiency²¹.” (line 43)

“The principal storyline here focuses on whether it is possible to achieve net-zero emissions across the cement and concrete cycle without relying on CCS, which belongs to the lowest tier of mitigation strategies²¹.” (line 74)

In addition, we have summarized the barriers to supply- and demand-side strategy implementation in a table and amended the discussion.

Table S28 Barriers in the implementation of several supply- and demand-side strategies. Based on a critical review of 37 previous studies [64], those mentioned more than once are checked.

Energy efficiency improvement	Low-carbon fuel utilization	Clinker-to-cement ratio reduction	Low-carbon cement chemistries	CCUS	Material efficiency improvement
-----------------------------	-----------------------------------	-------------------------------	------	---------------------------------

Economy	Higher cost	✓	✓	✓	✓	✓	✓	
	Availability of materials		✓	✓	✓			
	Market uncertainty					✓		
	Lack of demand	✓		✓	✓	✓	✓	
	Market acceptance			✓	✓			
Technical	Fragmented supply chain						✓	
	Lack of infrastructure		✓			✓		
	Lack of expertise	✓						
	Poor information	✓					✓	
	Additional energy requirements					✓		
	Time constraints						✓	
	Longer curing time			✓	✓			
	Structural strength			✓				
	Regulatory	Regulatory requirements		✓				✓
		Complex bureaucracy		✓			✓	
Risk concern							✓	
Lack of effective policies				✓		✓	✓	
Institutional challenges		✓		✓				
Social	Social acceptance		✓			✓		
	Industry culture						✓	

Comment

- Time horizon. Although 2050 was chosen to align with the international climate targets, are there insurmountable barriers that would prevent the Japanese concrete sector to reach carbon neutrality before 2050? If the authors were in control of the policies, how fast would they decarbonize the sector?

>Response

Thank you for emphasizing this important point. We admit that this study does not answer the question of how fast the concrete sector can be decarbonized. In practice, the speed of diffusion of each strategy is influenced by multiple factors, including political and regulatory procedures, infrastructure development, and information accumulation. One study examining the speed of different technological and social transitions suggested that while there are certain similarities in the patterns of transition progress, speed varies markedly and on a case-by-case basis. Therefore, the scope of

this paper is limited to whether decarbonization is achievable by 2050 based on simplistic strategy implementation assumptions. This point is now emphasized by adding the following text:

“Although the speed of diffusion of each strategy is influenced by multiple factors, including political and regulatory procedures, infrastructure development, and information accumulation, for simplicity, the speed of implementation is modeled assuming linear growth from 2021 to 2050 ³⁹.” (line 460)

Comment

• Supply-side strategies

o Low-carbon fuel utilization. Why the use of biomass and hydrogen was not considered? How were the emissions from natural gas (including leakage) and waste fuels modelled? What would be the fate of waste if not used for cement? Please report the main emission factors used for the study in the text.

>Response

Thank you for the question regarding this point. The breakdown of low-carbon fuels follows the industry roadmap because the conventional supply-side strategy to which low-carbon fuel utilization belongs is supposed to enhance the current industry's efforts to reduce CO₂ emissions. Emission factors for natural gas and waste fuels are based on national official values published by the government, and are presented in Table S12. In this case, the emission factors for natural gas do not take into account leakage during the extraction, treatment, transportation, storage, and other processes (CO₂ emissions from fuel leakage in Japan account for about 0.1% of total emissions). Basically, to avoid making the main text too long, details on technologies and parameters are presented in the Supplementary Information. This point was emphasized by adding following text:

“We consider a total of 16 strategies to achieve net-zero emissions in the cement and concrete sector. These can be broadly classified into three categories: conventional supply-side strategies (six strategies), emerging supply-side strategies (three strategies), and demand-side strategies (seven strategies). A brief description of each strategy group is given below, with detailed technical descriptions, assumed parameters, and barriers to implementation to be found in Section 3 of the Supplementary Information.” (line 446)

Comment

o Clinker-to-cement ratio reduction.

o Why only a 40% replacement was considered? For instance, the use of calcined clay limestone cements could already achieve replacements of 50% [1].

o How was the clinker to SCM substitution ratio modelled? 1-to-1? The ratio changes depending on the strength and exposure conditions of concrete [2].

>Response

Thank you for your thought-provoking comments and helpful references. According to

your comments, we have modified our assumptions as follows:

“Since most of the CO₂ emitted during the cement production process comes from clinker production, reducing the clinker-to-cement ratio can contribute to emission savings. Clinker can be replaced by a variety of supplementary cementitious materials (SCMs), such as fly ash, ground granulated blast furnace slag (GGBFS), and calcined clay [35]. In Japan, GGBFS and fly ash are mainly standardized and used. However, the availability of these industrial by-products depends on steel production and coal-fired power generation, and their availability is expected to decrease in a net-zero future. Therefore, we focus on calcined clays, especially in combination with limestone (i.e., LC³ technology). Since the LC³-50 blend (50% clinker, 30% calcined clay, 15% limestone and 5% gypsum) has already proven to be capable of providing mechanical properties comparable to standard ordinary Portland cement as well as provide some durability improvements [36], this study adopts an aggressive target of 50% clinker replacement by 2050 [37]. Although the substitution ratio of SCMs to clinker can vary with compressive strength, workability, or exposure conditions, a one-to-one substitution ratio is assumed here, as in several existing studies [24,38].” (SI)

Comment

o From figure S17 there seems to be an abundance of slag compared to the use. However, from the reference reported ([48]), it looks like all the slag produced is already used. Why the discrepancy between the source and figure S17 (e.g., for 2017: total produced 23.03 kt, used 23.97)? Moreover, the amount of slag will probably reduce in the future thanks to fuel/technology changes in steel production [3]. How does this affect the results?

>Response

Thank you for this comment. The original figure showed only blast furnace slag used in the domestic cement industry. According to the statistical data, about half of the slag used in the cement industry is exported, which explains the large gap between the statistical and estimated values in the original graph. We have modified the figure caption to make this point clearer. In addition, given the limited availability of blast furnace slag in the net-zero future, we decided not to consider the use of blast furnace slag as a clinker substitute.

Fig. S29 Supply-demand balance of blast furnace slag in Japan when all strategies are fully implemented, 2000-2050. The gap between supply and demand includes use for exports, civil engineering, and construction. Data adapted from [66].

Comment

- o CCU. The role of CCU seems to play a crucial role in the abstract. However, not enough information is provided in the text regarding the technology and the emission accounting.
- o Does CCU refer only to concrete curing and mineralization? Is the CO₂ captured from the kiln?
- o Why CCU for non-concrete applications (e.g., fuel and chemical production) was not included?
- o How does CCU affect carbonation? I.e., CO₂-cured concrete will absorb the same amount of CO₂ in its lifetime than traditional concrete?

>Response

We thank the reviewer for emphasizing several important points regarding CCU technologies. The CCUs considered in this study are concrete curing and mineralization with CO₂ captured from the kiln. CCUs to non-concrete products, including fuels and chemicals, are not considered due to system boundary issues. If CCUs to other products are considered, the full life cycle of the other products must be added to the boundary or not receiving any emission credits in order to avoid double counting. To avoid such emission accounting complications, this study only considered CCUs related to concrete products. We assume that a total increase in CO₂ uptake of CO₂-cured concrete is approximately 12% over the service life of the concrete. These points are described as shown below. Principally, the details of the technology and parameters are provided in the Supplementally Information, but the following text has been added to the main manuscript to better clarify issues related to emission

accounting. To increase transparency in this domain, we have also made the code and dataset available in a public repository.

“CCU (concrete curing)

Concrete curing with CO₂ refers to an accelerated carbonation process in which CO₂ gas is injected more thoroughly during the batching and mixing of concrete or during the curing process of precast products. This technology can reduce the CO₂ emissions associated with the concrete cycle in two ways: increased CO₂ uptake through accelerated carbonation, and savings in binders to achieve the required compressive strength [43]. However, because of the risk of corroding the steel frame inside, the concrete produced using this technology is difficult to use for some applications such as for reinforced concrete buildings [44]. However, concrete produced in this way is well suited for use in exterior materials employed in the civil engineering field, such as paving blocks and fence foundations [45]. Accordingly, we assume a maximum application rate of 34%, based on the strength class data. In calculating emission savings, this study assumes the following [24]: (1) the energy penalty from CO₂ transport and injection is 0.3 kg-CO₂/t -concrete; (2) the total increase in CO₂ uptake is approximately 12% over the service life of the concrete; (3) binder reduction for both ready-mixed and precast concrete due to increased concrete compressive strength is approximately 13%.” (SI)

“CCU (mineralization)

Mineralization is a technology that absorbs CO₂ by exploiting the property of substances containing elements such as calcium and magnesium to become carbonate minerals upon contact with CO₂. The carbonates produced by this technology include calcium carbonate, potassium carbonate, and magnesium carbonate. Among these, calcium carbonate can be used as an aggregate for concrete, and a promising source of raw calcium is through recovery from industrial waste [44]. Therefore, we assume that 1% and 0.5% of concrete production will be able to use synthetic aggregates made from blast furnace slag and lime mud, respectively, by 2050 [24]. The use of fly ash and red mud is not considered, because coal-fired power generation is assumed to be phased out and Japan relies on imports for 100% of its aluminum ingots. Blast furnace slag should also decline as steel production is decarbonized, but the industry roadmap shows that some blast furnaces will remain in 2050 [46]. Further, Japan’s CCU roadmap specifies the utilization of blast furnace slag [45]. Consequently, we assume that the utilization of blast furnace slag for this technology is viable. In addition, we assume that 10% of demolished concrete can be converted into synthetic aggregates with this technology, with a 12% increase in CO₂ uptake over the service life of the concrete, which is similar to concrete curing [24]. The same energy penalty is assumed as for concrete curing [24]. Although the effect of the use of synthetic aggregates on the compressive strength of concrete is not well known, we assume that there is no particular effect.” (SI)

“As for CCU, there are two types of CCU to be considered: 'concrete curing', where CO₂ gas is injected during batching and mixing of concrete or the curing process of precast

products, and 'mineralization', where CO₂ is mineralizing with alkaline substances such as calcium and magnesium to form carbonate minerals. CCU technologies mitigate CO₂ emissions through increased CO₂ uptake and storage over the service life of concrete and reduced binder due to increased concrete compressive strength." (line 471)

Code: https://github.com/takumawatari/concrete_cycle_jp

Comment

o Line 266: What does it mean that CCU is excluded from national inventories? Emissions are accounted for the CO₂ user but not for the CO₂ producer? Please provide a reference and explain.

>Response

Thank you for this question. Our intent here is to show that any CO₂ uptake associated with concrete carbonation or CCU is not accounted for in the current Japanese national inventory. We have added references to clarify this point and changed the text as follows:

"Importantly, the decarbonization pathway prescribed here depends on how CO₂ uptake from concrete carbonation and CCUs is accounted for in the inventory. Currently, CO₂ uptake from either natural carbonation or CCUs is not accounted for in national inventories under the Paris Agreement⁴⁸. If this situation persists, the residual CO₂ emissions will need to be reduced in other ways. Since this can clearly make a substantial difference in the design of strategies and investments, there is an urgent need to start and settle the debate on how to account for concrete-related CO₂ uptake in national inventories under the Paris Agreement. If this is not done, the industry's vision of achieving net-zero emissions will be inconsistent with that at the national level under the Paris Agreement, which will in turn jeopardize the effectiveness of net-zero emissions in the context of specific climate goals." (line 311)

Comment

o Although I agree there is an urgent need for a clear methodology, it is clear what carbon neutral means: the concentration of CO₂ (and other greenhouse gases) in the atmosphere should not increase due to cement and concrete production.

>Response

Thank you for highlighting this important point. What we are calling for here is an agreed upon emissions accounting method. As mentioned above, CO₂ uptake by carbonation and CCUs is currently not taken into account in the Japanese national inventory. Therefore, we are concerned that if we do not quickly agree on an accounting method for concrete-related CO₂ uptake, industry pledges and actual national emissions under the Paris Agreement will not be consistent. The question of how it should be accounted for is clearly an important subject for future research. We have added following text to further emphasize this point:

"Overall, the message of this study is clear: cross-cutting strategies involving both the

supply and demand sides can decarbonize the entire Japanese cement and concrete cycle by 2050 without resorting to mass deployment of CCS. However, realizing these strategies will require (1) coordinated policies to raise awareness of the importance of demand-side strategies, (2) revised material flow-related targets consistent with national commitments to climate change, and (3) consensus in the accounting methods used to assess CO₂ uptake by carbonation and CCUs in national inventories.” (line 323)

Comment

- Demand-side strategies. More information should be provided regarding the implementation of these strategies.

- o What are the barriers for each strategy?

>Response

Thank you for this useful comment. We have added the following discussion and data regarding barriers to demand-side strategies.

“Despite the importance of demand-side strategies, there is absolutely no incentive for the materials industry to promote efficient material use, since the current profits are directly related to the volume of materials sold. More importantly, purchases of cement and concrete currently account for less than 10% of total spending in the building and infrastructure construction sector, suggesting the weak economic incentives to pursue material efficiency even on the demand side (**Fig. S31-33** in the Supplementary Information). These facts suggest that material efficiency gains will not progress spontaneously, but require clear policy guidance. Since demand-side strategies, especially those involving material-efficient design and more intensive use, have been rarely covered by existing policies, adjustments will be needed to raise awareness of the importance of material efficiency, which is much less recognized than energy efficiency^{43,44}.” (line 284)

Fig. S31 Cement and concrete purchases as a percentage of total expenditures in the residential building construction sector. Data adapted from the Japanese Input-Output table [68].

Comment

o Why a gradual implementation curve was considered for more intensive use? I.e., what is the issue with abrupt changes?

>Response

Thank you for this question. This is because assuming linear improvements to more intensive uses can result in unrealistic changes to new construction, and thus concrete demand. By “unrealistic”, we mean something that greatly exceeds the range of rates of change observed in the past. Therefore, we assume that it is reasonable to use a gradual implementation curve, as has been done in several previous studies:

1. Fishman, T.; Heeren, N.; Pauliuk, S.; Berrill, P.; Tu, Q.; Wolfram, P.; Hertwich, E. A comprehensive set of global scenarios of housing, mobility, and material efficiency for material cycles and energy systems modelling. 2020, 1–16.
2. Wolfram, P.; Tu, Q.; Heeren, N.; Pauliuk, S.; Hertwich, E.G. Material efficiency and climate change mitigation of passenger vehicles. J. Ind. Ecol. 2020, 1–17.

The following graphs have been added to make the assumptions clearer.

Fig. S19 Building stock per capita in Japan, 2020-2050.

Fig. S20 Infrastructure stock per capita in Japan, 1950-2020.

Comment

o How is the lifetime extension modelled? How does refurbishment/maintenance affect the greenhouse gas balance? How is it ensured a longer lifetime?

>Response

Thank you for this question. We have revised the text regarding the lifetime extension as follows:

“Extending the service life of buildings and infrastructure can curb demand for new construction activities. In Japan, the 'Act for Promotion of Long-Life Quality Housing' was enacted in 2008 to encourage the extension of the service life of housing through economic incentives. Further, ministries and agencies have formulated action plans for extending the service life of infrastructure facilities under their jurisdiction based on the 'Basic Plan for Extending the Service Life of Infrastructure'. Key measures include early detection of deterioration and damage through periodic inspections,

development of electronic maintenance information, and use of sensors and robots in inspections and repairs. Based on the literature [57,58], this study assumes that the service life of buildings and infrastructure can be extended by 90% and 30% from current levels, respectively. Relatively conservative assumptions are made that the lifetime buildings and infrastructure that are constructed after 2021 will be extended, which means that the lifetime extension of existing stock is not considered. Increased maintenance for extended lifetime may involve additional carbon emissions, but that is not considered here due to lack of reliable data.” (SI)

Comment

o How is the efficient design strategy implementation modelled?

>Response

Thank you for this question. Material-efficient design is reflected in two variables: concrete intensity of buildings and infrastructure, and the cement content of concrete. The description has been revised as follows to highlight this point:

“Performance-based design allows architects and contractors to design concrete mixtures that meet the necessary mechanical and durability specifications with less cement. Some evidence suggests that the cement content of concrete could be reduced by 15-20% without compromising compressive strength [47,48]. Importantly, a previous study showed that prescriptive design, which specifies the allowable cementitious content in concrete, induces over-use of cement [49]. According to their estimates, using performance-based design rather than prescriptive design could reduce carbon emissions per required volume of concrete by up to 30%. Here, we assume that the cement content of concrete can be reduced by 15% by 2050 [38]. Furthermore, precast components allow designers to manufacture concrete components with greater precision and reliability using less cement. Post-tensioning techniques can make parts of concrete elements thinner by stressing the rebar in the concrete floor slab before applying external loads. Changes in the way elements are specified could reduce the amount of cementitious materials used in structural components, especially as overdesign is often employed by designers when constructing structural elements. Together, we assume that a 13% reduction in the concrete intensity of concrete structures can be achieved without interfering with its function by 2050 [24,38]. Currently, the concrete intensity of buildings in Japan is two to four times higher than in the U.S. and China. This difference is mainly due to the structure of the building, especially the thickness of the foundation, piles, and columns [50]. Since the strategies considered here do not compromise strength or durability of structures, we assume that transitions can be encouraged by making adjustments to building codes and through appropriate code enforcement.” (SI)

Comment

• Carbonation.

o Is concrete assumed to be pulverized at its end of life to optimize the carbonation? If yes, how does the energy consumption affect the CO₂ balance? If concrete was landfilled, how much CO₂ would be absorbed?

o Line 100: How was the CO₂ uptake in 2019 calculated? Is it just from in-use stock or from waste concrete?

o Line 340: please specify the range of carbonation assumed for the different concrete types: e.g., 10%-15% of the calcination emissions are reabsorbed during the service life of concrete, additional 10-15% at the end of life.

>Response

Thank you for highlighting several important points regarding carbonation. Firstly, CO₂ uptake by carbonation is estimated by a physicochemical model that considers uptake from in-use stock, demolition waste, construction waste, and cement kiln dust. Crushing of concrete structures can generate CO₂ due to the use of heavy machinery, but this effect is not considered in this study as it is difficult to assign them to a single material. To better clarify the points you raised, we have added the following sentences:

“The emission sources considered in this study are roughly divided into six major categories: cement production, virgin aggregate production, recycled aggregate production, concrete mixing and batching, concrete on-site placement, and transportation activities. Emissions from the use and dismantling phases are excluded from the model as it is difficult to assign them to a single material.” (line 376)

“The concrete carbonation at the demolition stage is modeled by assuming that the particles of demolition waste are spherical, since concrete structures are usually crushed into small pieces in order to recycle steel and facilitate subsequent transport of demolition waste. Thus, the carbonated fraction of demolished concrete is estimated as follows:” (SI)

“The CO₂ uptake by concrete carbonation reached 8.2 Mt-CO₂ (7.1-10.0 Mt-CO₂ interquartile range) in 2019, equivalent to approximately 24% of concrete-related emissions and approximately 1% of Japan’s total emissions. In-use stocks act as the largest sink during the service life of buildings and/or infrastructure, absorbing approximately 74% of the total uptake. Another 22% is derived from demolition waste at the end-of-life stage, and the remaining 4% from cement kiln dust and construction waste (Fig. S26 in the Supplementary Information).” (line 101)

Fig. S26 CO₂ uptake from concrete carbonation, 1950-2019.

Comment

- Applicability.

o What policies should be brought forward to reach carbon neutrality according to the authors? From the work, it looks like more intensive use and efficient design are the most efficient options. How can they be implemented?

>Response

We thank the reviewer for emphasizing this extremely important point. We have added text to the discussion describing specific policies that can be adopted to advance demand-side strategies.

“Since demand-side strategies, especially those involving material-efficient design and more intensive use, have been rarely covered by existing policies, adjustments will be needed to raise awareness of the importance of material efficiency, which is much less recognized than energy efficiency^{43,44}. For example, as a condition for tax incentives, the ‘Low Carbon Building Certification Scheme’⁴⁵ only certifies buildings that are wooden, long-lived, or made of cement containing blast furnace slag or fly ash. Adding standards for cement and concrete usage, along with adjustments to building codes, could be an effective way to encourage material-efficient design. Another possibility is to amend the ‘Building Energy Efficiency Act’⁴⁶, which regulates compliance with energy efficiency standards, and the ‘Low Carbon City Development Plan’⁴⁵, which provides subsidies to selected municipalities. Since the current focus of these laws is primarily on operational energy use, articulating “embodied” carbon can add compelling reasons for stakeholders to build materially efficient buildings and urban structures. The newly enacted subsidy program for life-cycle carbon-minus houses is innovative in that it includes embodied carbon in the assessment⁴⁷. However, since only the use of blast furnace slag-containing cement and lifetime extension are explicitly mentioned in the material-related efforts, adjustments will need to be made

to encourage more diversified demand-side strategies.” (line 291)

Comment

o Line 248: please specify the material flow-related indicators that you would place as national targets

>Response

We apologize for our earlier manuscript not making this point clear. The following text has been added:

“Our analysis bridges this gap by providing a set of indicators, i.e., domestic consumption and in-use stock, that are fully consistent with the net-zero emissions target.” (line 265)

Comment

Other questions:

• Line 123: Does that mean that the amount of waste concrete equals the amount of new concrete?

>Response

Yes, to some extent, but some of it is not. Not all concrete structures that have reached the end of their useful life become waste; some become hibernating stock. Therefore, the statement here implies that the amount of new concrete structures being added to society is roughly equal to the amount of concrete structures that are no longer in use. Although there is little reliable information worldwide on hibernating stock, we have made every effort to use reasonable assumptions wherever possible. We have added comparisons with other high-income countries to reinforce the trend of stock saturation observed in Japan.

Fig. S24 International comparison of in-use stock of cement equivalent, 1950-2020. Data for countries other than Japan are based on our previous study [15], and country

selection is based on other material flow analysis studies [65].

Comment

- Line 190: How could the resource use decoupling be implemented in countries that did not reach stock saturation?

>Response

Indeed, this is an important point. We have added text to the discussion to describe the implications for countries other than Japan.

“Overall, the message of this study is clear: cross-cutting strategies involving both the supply and demand sides can decarbonize the entire Japanese cement and concrete cycle by 2050 without resorting to mass deployment of CCS. However, realizing these strategies will require (1) coordinated policies to raise awareness of the importance of demand-side strategies, (2) revised material flow-related targets consistent with national commitments to climate change, and (3) consensus in the accounting methods used to assess CO₂ uptake by carbonation and CCUs in national inventories. These perspectives are not unique to Japan, given that most high-income countries in the world trace similar patterns of cement and concrete use. In particular, many middle- and low-income countries, which are in the process of expanding their material stocks^{49,50}, now have an important opportunity to systematically converge their stock growth at a much lower level than the current developed countries by incorporating efficient use of cement and concrete into urban development planning.” (line 323)

Comment

- Line 304: What's the difference between steel-reinforced concrete and reinforced concrete?

>Response

Reinforced concrete refers to a structure built by pouring concrete into a formwork made of a mesh of reinforcing bars. Steel-reinforced concrete, on the other hand, is a structure in which a steel frame is contained within reinforced concrete columns and beams. The process is almost identical to that of reinforced concrete construction, but elements of steel construction are incorporated. Such structures are used primarily for high-rise buildings and large-scale construction.

Comment

- Fig 4: Is carbonation included in the figure? How?

>Response

Yes, Fig 4 includes carbon uptake by concrete carbonation, which is calculated by a physicochemical model. We have added Figure 5 to better clarify this point.

Comment

- Supp. 5. Why cementitious solidifiers were not included in the assessment if they represent a significant share of cement use?

>Response

Thank you for this comment. The reason why cementitious solidifiers are not considered in this study is due to the model structure and incomplete data. Our model connects buildings and infrastructure construction activity information to the cement and concrete cycle. This process requires material intensity data for each construction activity, but cement used as cementitious solidifiers are not included in the data maintained by the government. Moreover, data on their long-term production, disposal, and carbonation parameters are not well developed enough to model them. Addressing this limitation is obviously important, but we are currently struggling to address this issue.

Comment

Typos and small edits:

- Line 36: I disagree that “the question regarding what actions need to be taken to decarbonize the cement sector is largely unexplored”, as it is proven by the following paragraph.

>Response

We agree and have deleted the words “largely unexplored”.

Comment

- Line 62: avoid the use of non-scientific terms as “huge”
- Line 206: after better a verb is missing

>Response

Fixed. Thank you for pointing these out!

[1] K. Scrivener, F. Martirena, S. Bishnoi, S. Maity, Calcined clay limestone cements (LC3), *Cem. Concr. Res.* 114 (2018) 49–56. <https://doi.org/10.1016/j.cemconres.2017.08.017>.

[2] A. Arrigoni, D.K. Panesar, M. Duhamel, T. Opher, S. Saxe, I.D. Posen, H.L. MacLean, Life cycle greenhouse gas emissions of concrete containing supplementary cementitious materials: cut-off vs. substitution, *J. Clean. Prod.* 263 (2020) 121465. <https://doi.org/https://doi.org/10.1016/j.jclepro.2020.121465>.

[3] International Energy Agency, Cement Sustainability Initiative, Technology Roadmap. Low-Carbon Transition in the Cement Industry, Paris, France, 2018. https://doi.org/10.1007/SpringerReference_7300.

Response to reviewers, second round review –

Reviewer #1 (Remarks to the Author):

The authors have addressed all of my concerns.

Reviewer #2 (Remarks to the Author):

Thanks for corrections.

I would disagree on one point, carbonation from stock pile.

In order to have fast re-carbonation of crushed aggregates, a stock pile is not sufficient as CO₂ access to inner part of the stock pile is not direct. It follows diffusion process which takes time.

Ref: Thiery et al., 2013. Carbonation kinetics of a bed recycled concrete aggregates: a laboratory study on model materials. Cement and concrete research, 46, 50-65.

In order to gain carbon uptake from crushed aggregates, an active system needs to be implemented.

Reviewer #3 (Remarks to the Author):

I would like to thank the authors for addressing my comments and improving the quality of the manuscript. There are still a few points that in my opinion should be better clarified before publication though:

- 1) It is still unclear to me how the sector can reach carbon neutrality if not all of the CO₂ emitted during calcination is reabsorbed by the concrete? If this is the case, please make it clear.
- 2) Emissions from natural gas life cycle are much more than 0.1% as recent studies showed (e.g., methane tracker from IEA). At least a comment should be made that if cement production is "cleaned" via natural gas, leakage throughout the natural gas supply chain should be minimized.
- 3) Please provide more information regarding the trend of figure S26: why is the CO₂ uptake decreasing?
- 4) Regarding policies including embodied carbon, how would you measure it? Per square meter? Or per cubic meter of concrete? A building could use very sustainable concrete, but too much of it.
- 5) Please mention in the manuscript (or SM) why cementitious solidifiers were not included in the assessment (i.e., the answer you provided).

Response to reviewers

Reviewer #1 (Remarks to the Author):

Comment

The authors have addressed all of my concerns.

>Response

Thank you!

Reviewer #2 (Remarks to the Author):

Comment

Thanks for corrections.

I would disagree on one point, carbonation from stock pile. In order to have fast re-carbonation of crushed aggregates, a stock pile is not sufficient as CO₂ access to inner part of the stock pile is not direct. It follows diffusion process which takes time.

Ref: Thiery et al., 2013. Carbonation kinetics of a bed recycled concrete aggregates: a laboratory study on model materials. *Cement and concrete research*, 46, 50-65.

In order to gain carbon uptake from crushed aggregates, an active system needs to be implemented.

>Response

We are very grateful to the reviewer for this deeply insightful comment. Certainly, we have not considered active systems that promote faster carbonation of demolished concrete, but simply calculated the effect of longer stockpiling periods on CO₂ uptake according to Fick's laws of diffusion. We have modified the text to clarify this point as follows:

“During demolition, concrete is crushed into small pieces, which increases the surface area and promotes the carbonation process. Crushed concrete pieces are stored in intermediate treatment facilities for an average of 0.2 years in Japan [33]. Extending this storage period will promote the CO₂ absorption process of demolished concrete since the surface area of crushed concrete exposed to air is limited in a landfill environment [63]. The progress of carbonation of demolished concrete is estimated according to Fick's law of diffusion. In this case, we simply assume a longer stockpiling period and do not consider active systems, such as particle size refinement or wet-dry cycles that promote faster carbonation. Carbonation of demolished concrete proceeds faster with smaller particle sizes, as the area in contact with air is more extensive [64]. In addition, repeated drying and wetting promote faster carbonation due to water intervention [65]. While particle size refinement or wet-dry cycles can expedite carbonation, a key barrier to adopting these two measures at a large scale is the cost of transporting crushed concrete pieces to treatment facilities [24]. Compared to the aforementioned active measures, extending the stockpiling duration entails less transportation, but the maximum length of time that crushed concrete pieces can stay stockpiled is dependent on social acceptance and local regulations concrete. For example, the EU Construction and Demolition Waste Management Protocol and Guidelines suggest that the maximum stockpiling time is limited to one year [24]. Therefore, this study assumes that the stockpiling period of demolition waste can be extended to one year by 2050.”

Reviewer #3 (Remarks to the Author):

Comment

I would like to thank the authors for addressing my comments and improving the quality of the manuscript. There are still a few points that in my opinion should be better clarified before publication though:

>Response

Thank you for your incisive comments, which were very helpful for us in improving the manuscript. We hope that the following revisions clarify the issues that you have raised.

Comment

1) It is still unclear to me how the sector can reach carbon neutrality if not all of the CO₂ emitted during calcination is reabsorbed by the concrete? If this is the case, please make it clear.

>Response

Thank you very much for emphasizing this point and for the opportunity to clarify this issue. What we mean here by 'carbon neutrality' is that emissions from production activities are consistent with system-wide uptake and storage, including from in-use stock and demolition waste. Since much of the in-use concrete produced by past production has not yet been fully carbonated, the CO₂ emissions from production activities in 2050 are balanced by CO₂ uptake mainly from the in-use stock and demolition waste. To better clarify this point, we have added the following text and have modified Figure 4.

"It is important to note that net-zero emissions, as defined here, refer to a state of equal CO₂ emissions and uptake, both of which change depending on strategy implementation (Fig. 4). With supply-side strategies alone, CO₂ emissions in 2050 (11 Mt-CO₂) far exceed CO₂ uptake and storage by natural carbonation and CCUs (-7 Mt-CO₂), resulting in net positive CO₂ fluxes in 2050 (4 Mt-CO₂). With both supply- and demand-side strategies in place, 2050 CO₂ emissions could be reduced more deeply (4 Mt-CO₂). Thus, the residual emissions are fully offset by CO₂ uptake and storage (-5 Mt-CO₂), providing net negative CO₂ fluxes (-1 Mt-CO₂) across the entire cement and concrete cycle. This condition refers to a state in which emissions associated with production activities are balanced by system-wide uptake and storage, mainly from in-use stock and demolition waste brought about by past cement and concrete production."

Fig. 4 CO₂ emission and uptake associated with cement and concrete cycle in Japan under the three representative scenarios, 2010–2050. The vertical dashed lines mark the year in which the future scenarios begin (2020).

Comment

2) Emissions from natural gas life cycle are much more than 0.1% as recent studies showed (e.g., methane tracker from IEA). At least a comment should be made that if cement production is "cleaned" via natural gas, leakage throughout the natural gas supply chain should be minimized.

>Response

We thank the reviewer for emphasizing this point. Certainly, our previous manuscripts were unclear in this regard and we have added the following text to better clarify this point:

“The main fuel used to fire kilns has historically been coal, due to its widespread availability, high calorific value, and generally low cost [7]. The Japanese cement industry has a long-term vision to completely switch to natural gas and waste fuels in order to reduce CO₂ emissions from fuel combustion [34]. This study assumes that the shares of natural gas and waste fuels (assuming waste plastics) in thermal energy generation can be increased from current levels to 50% each by 2050. This means that there will be a complete shift away from coal in fuel combustion by 2050. The emission factor for each fuel follows the national emission factor shown in **Table S12**. Note that the emission factor only accounts for emissions associated with fuel combustion and does not include leakage from fossil fuel mining, processing, transportation, storage, and other processes. Currently, due to the small scale of domestic production in Japan, emissions from fuel leakage in the natural gas supply chain account for only about 0.1% of total emissions [35], but this figure increases significantly when imports are taken into account [36]. Therefore, when expanding the use of natural gas in cement production, it is important to minimize fuel leakage

throughout the natural gas supply chain.”

Comment

3) Please provide more information regarding the trend of figure S26: why is the CO₂ uptake decreasing?

>Response

Thank you for this comment. The reason why CO₂ uptake has decreased over the last 30 years is mainly due to the utilization of in-use mortars. When cement is used as mortar, carbonation proceeds faster than in concrete due to its formulation and because it has a larger contact area with the atmosphere than concrete. Therefore, uptake from in-use mortars is strongly linked to the cement production trend, which has led to a downward trend in total uptake. Such trends are generally consistent with observations in previous studies. We have added the following explanation to clarify this point.

Fig. S26 CO₂ uptake from concrete carbonation, 1950-2019.

Note: CO₂ uptake has decreased for the last 30 years, in line with cement production trends. Such linkage is mainly due to in-use mortars. When cement is used as mortar, carbonation proceeds faster than in concrete due to its formulation and because it has a larger contact area with the atmosphere than concrete. Therefore, uptake from in-use mortars is strongly linked to the cement production trend, which has led to a downward trend in total uptake. Such trends are generally consistent with observations in previous studies [4,5].

Comment

4) Regarding policies including embodied carbon, how would you measure it? Per square meter? Or per cubic meter of concrete? A building could use very sustainable concrete, but too much of it.

>Response

Thank you for this constructive comment, as this is indeed an important point. We have added the following text to better clarify this issue:

“Another possibility is to amend the ‘Building Energy Efficiency Act’⁴⁶, which regulates compliance with energy efficiency standards, and the ‘Low Carbon City Development Plan’⁴⁵, which provides subsidies to selected municipalities. Since the current focus of these laws is primarily on operational energy use, articulating “embodied” carbon per specific functional unit (e.g., unit floor area) can add compelling reasons for stakeholders to build materially efficient buildings and urban structures. The newly enacted subsidy program for life-cycle carbon-minus houses is innovative in that it includes embodied carbon in the assessment⁴⁸. However, since only the use of blast furnace slag-containing cement and lifetime extension are explicitly mentioned in the material-related efforts, adjustments will need to be made to encourage more diversified demand-side strategies. The key here is to recognize that the amount of cement and concrete that can be produced and used in a net-zero future is not infinite, and that it is considerably less than current levels. This means that stock growth and turnover must be managed simultaneously so as not to induce excessive use of low or zero carbon materials⁴⁷.”

Comment

5) Please mention in the manuscript (or SM) why cementitious solidifiers were not included in the assessment (i.e., the answer you provided).

>Response

Thank you for highlighting this point. We have added the following text:

“This section provides a validation of our estimated results (**Fig. S34**). We can confirm that estimates of domestic cement consumption in this study are generally in good agreement with publicly available statistics. On the other hand, the estimates for cement and concrete stocks show some deviation from the two previous studies. This could be explained simply by the difference in the applications considered.

First, Cao et al. [15] considered all cement applications, including cementitious solidifiers, which are not covered in this study. The reason why cementitious solidifiers are not considered in this study is due to the model structure and incomplete data. Our model connects buildings and infrastructure construction activity information to the cement and concrete cycle. This process requires material intensity data for each construction activity, but the cement that is used as cementitious solidifiers is not included in the data collated by the government. We can assume that this is the prime reason for the larger estimates compared to those obtained in this study. Indeed, despite the difference in absolute values, the long-term trend itself is in good

agreement with their findings, with stock growth becoming saturated from around 2000.

In contrast, Tanikawa et al. [29] estimated concrete stock volumes smaller than that estimated in this study, which may also reflect the difference in the applications considered. Specifically, the estimate of Tanikawa et al. [29] excludes landslide and flood control measures, agriculture, forestry and fisheries, and waste treatment. These applications account for approximately 20% of the total stock, which precisely explains the difference in the estimates.”

Response to reviewers, third round review –

REVIEWERS' COMMENTS

Reviewer #3 (Remarks to the Author):

Thank you for addressing all my comments and congratulations for your work!